# *TSENOR*: Highly-Efficient Algorithm for Finding Transposable N:M Sparse Masks

**Xiang Meng**
Operations Research Center
Massachusetts Institute of Technology
`mengx@mit.edu`

**Mehdi Makni**
Operations Research Center
Massachusetts Institute of Technology
`mmakni@mit.edu`

**Rahul Mazumder**
Operations Research Center
Massachusetts Institute of Technology
`rahulmaz@mit.edu`

## Abstract

Network pruning reduces computational requirements of large neural networks, with N:M sparsity—retaining only N out of every M consecutive weights—offering a compelling balance between compressed model quality and hardware acceleration. However, N:M sparsity only accelerates forward-pass computations, as N:M patterns are not preserved during matrix transposition, limiting efficiency during training where both passes are computationally intensive. While transposable N:M sparsity has been proposed to address this limitation, existing methods for finding transposable N:M sparse masks either fail to scale to large models or are restricted to M=4 which results in suboptimal compression-accuracy trade-off. We introduce an efficient solver for transposable N:M masks that scales to billion-parameter models. We formulate mask generation as optimal transport problems and solve through entropy regularization and Dykstra's algorithm, followed by a rounding procedure. Our tensor-based implementation exploits GPU parallelism, achieving up to 100× speedup with only 1-10% error compared to existing methods. Our approach can be integrated with layer-wise N:M pruning frameworks including Wanda, SparseGPT and ALPS to produce transposable N:M sparse models with arbitrary N:M values. Experiments show that LLaMA3.2-8B with transposable 16:32 sparsity maintains performance close to its standard N:M counterpart and outperforms standard 2:4 sparse model, showing the practical value of our approach. Our code is available at `https://github.com/mazumder-lab/TSENOR`.

## 1 Introduction

Deep neural networks (DNNs) have become ubiquitous across multiple fields, notably in natural language processing, speech recognition, and autonomous systems [Brown et al., 2020, Radford et al., 2023, Chen et al., 2024]. While these models achieve remarkable performance, they come with substantial computational and storage requirements. State-of-the-art models, such as GPT-4 Achiam et al. [2023] and LLaMA3 Dubey et al. [2024], contain hundreds of billions of parameters, necessitating multiple high-end GPUs for both training and inference. This massive scale poses significant challenges for real-world deployment and increases energy consumption [Wu et al., 2022].

To address these challenges, various model compression techniques have been proposed, including quantization [Lin et al., 2023b, Dettmers et al., 2023], knowledge distillation [Gou et al., 2021], and network pruning [Han et al., 2015, Cheng et al., 2024]. Network pruning reduces model size

by eliminating redundant or less important weights and can be categorized into unstructured and structured approaches. Unstructured pruning [Benbaki et al., 2023, Frantar and Alistarh, 2023] and related variants Meng et al. [2024b] removes individual weights regardless of their position, achieving high compression ratios with minimal accuracy degradation. However, it often fails to deliver practical acceleration on commodity hardware due to irregular memory access patterns and the overhead of encoding sparse structures [Zhu et al., 2019, Tang et al., 2021]. In contrast, structured pruning [Wen et al., 2016, He et al., 2017, Meng et al.] removes entire structural components (channels, filters, heads) and is more hardware-friendly, but typically results in greater accuracy loss at high sparsity levels.

N:M sparsity—where only N out of every M consecutive weights are retained—offers a compelling middle ground. This fine-grained structured sparsity maintains unstructured pruning benefits while enabling hardware acceleration [Nvidia, 2020, Lin et al., 2023a]. Despite advances in N:M sparsity methods [Mishra et al., 2021, Zhou et al., 2021, Lu et al., 2023, Sun et al., 2023, Meng et al., 2024a, Bambhaniya et al., 2024, Lucas and Mazumder], a critical limitation persists: they accelerate only forward-pass operations ($Y = WX$) but not backward-pass operations ($\partial L/\partial X = W^T \cdot \partial L/\partial Y$), as N:M sparsity patterns are not preserved during matrix transposition. Consequently, N:M sparsity only provide partial acceleration during training, where both forward and backward passes are computationally intensive.

Transposable N:M sparsity addresses this by designing patterns that maintain N:M structure in both a matrix and its transpose. Despite its usefulness and promise, finding transposable N:M sparse masks presents significant algorithmic challenges. Due to a lack of efficient algorithms, the full potential of transposable N:M sparsity perhaps remains to be realized—a key motivation of our work. Interesting prior works include: Hubara et al. [2021a] compute the transposable mask via minimum-cost flow, which is computationally expensive and does not scale to LLMs with billions of parameters; Hu et al. [2024] solve the special case of transposable 2:4 masks for transformers efficiently through exhaustive search, but their method cannot be generalized to larger M values ($\geq 8$) as the size of search space grows exponentially fast. Transposable 2:4 sparsity imposes strong structural constraints that can adversely affect model accuracy compared to larger N:M values.

In this paper, we introduce *TSENOR*[1], a novel algorithmic framework for finding binary masks with transposable N:M sparsity that scales to LLMs. We formulate optimal mask selection as an integer program and observe a novel connection to optimal transport. Leveraging this connection, we solve the relaxed problem using entropy regularization and Dykstra's algorithm. This is followed by a new rounding procedure to recover binary masks. Our approach efficiently handles arbitrary N:M patterns, which is crucial as larger M values significantly reduce performance degradation—especially when we compare non-transposable to transposable sparsity. Our method's tensor-based implementation enables seamless integration with existing layerwise pruning frameworks—this accelerates both forward and backward passes while preserving model quality. Our contributions include:

1. We formulate transposable N:M sparse mask generation as multiple optimal transport problems with capacity constraints, and solve them simultaneously by entropy regularization and Dykstra's algorithm. This approach is highly parallelizable and efficiently handles arbitrary N:M patterns.

2. The solutions obtained from Dykstra's algorithm are fractional and cannot be directly used as binary masks. To this end, we propose a GPU-optimized rounding procedure that converts fractional solutions to high-quality binary masks through greedy selection and local search. Our tensor-based implementation can process millions of blocks simultaneously, achieving up to $10^3$ times speedup over vanilla approach.

3. We show the integration of *TSENOR* with existing layer-wise N:M pruning frameworks to generate transposable N:M sparse networks. Specifically, we incorporate it as a plug-in procedure into leading pruning approaches such as Wanda [Sun et al., 2023], SparseGPT [Frantar and Alistarh, 2023], and ALPS [Meng et al., 2024a]. For ALPS, we provide novel convergence guarantees for the resulting framework.

4. Experimental results show that our method generates masks with 1-10% less relative error compared to existing heuristics and runs up to $10^2$ times faster. We demonstrate that transitioning from non-transposable to transposable sparsity with larger M values (16:32) results in only 12% of the performance loss compared to smaller M values (2:4). Moreover, models with transposable

---

[1]**T**ransposable N:M **S**parsity with **EN**tropy regularization and **O**ptimized **R**ounding

16:32 sparsity outperform those with non-transposable 2:4 sparsity, confirming the importance of our efficient solver for arbitrary (especially large M) transposable patterns. Our code is publicly available at: `https://github.com/mazumder-lab/TSENOR`.

## 2 Preliminaries and related work

**Network pruning with N:M sparsity** Techniques for obtaining N:M sparse networks fall into four categories: (i) Sparse training methods that mitigate performance degradation induced by N:M sparsity through carefully designed training strategies [Mishra et al., 2021, Zhou et al., 2021, Lu et al., 2023, Bambhaniya et al., 2024]; (ii) Pruning approaches that identify high-quality sparse masks for given N:M patterns by minimizing layerwise reconstruction error on calibration samples [Frantar and Alistarh, 2023, Meng et al., 2024a] or estimating weight importance [Sun et al., 2023]; (iii) Configuration search methods that determine layer-specific N:M sparsity patterns [Sun et al., 2021, Huang et al., 2024]; and (iv) Methods performing channel permutation on the weight matrix to improve pruned N:M sparse model performance [Pool and Yu, 2021, Mahajan et al., 2024].

**Transposable N:M sparsity** Transposable N:M sparsity remains relatively unexplored compared to its non-transposable counterpart. Hubara et al. [2021a] first proposed transposable N:M masks to accelerate both forward and backward passes during training, introducing the minimum-cost flow method as well as a greedy heuristic for mask generation. Hu et al. [2024] developed a highly efficient vectorized approach specifically for transposable 2:4 mask generation. Alternatively, Zhang et al. [2023] proposed training non-transposable N:M sparse networks using gradients approximated by applying transposable N:M sparse masks during the backward pass.

**Hardware implementation of N:M sparse networks** NVIDIA's Sparse Tensor Cores in the Ampere GPU architecture Nvidia [2020] support 2:4 sparsity acceleration. Castro et al. [2023] introduces Spatha sparse library, enabling arbitrary N:M patterns on Sparse Tensor Cores. Alternative approaches like nmSPARSE [Lin et al., 2023a] and NM-SPMM [Ma et al., 2025] design GPU kernels with memory access optimization and blocking mechanism to support arbitrary N:M patterns without requiring specialized hardware. Fang et al. [2023] designed computation-efficient training through algorithm-architecture-dataflow co-design, while Liu et al. [2025] proposed transposable block-wise N:M sparsity with dedicated tensor cores.

**GPU-accelerated optimization** Our approach formulates transposable N:M mask generation as solving millions of small-scale optimal transport problems, efficiently parallelized using GPU acceleration. In contrast, most existing GPU-accelerated optimization methods focus on solving a single large-scale problem. Examples include efficient GPU implementations of linear programming solvers [Lu and Yang, 2023, Pacaud et al., 2024] and parallelizable approaches for large-scale optimal transport problems [Cuturi, 2013, Mai et al., 2021].

## 3 Computing high-quality transposable N:M mask efficiently

Given a weight matrix $\mathbf{W}$, the core problem in transposable N:M sparsity is to determine a binary mask that maximally preserves the magnitude of weights in $\mathbf{W}$ under the transposable N:M constraint. This can be formulated as:

$$\max_{\mathbf{S}} \sum_{i,j} \mathbf{S}_{ij}|\mathbf{W}_{ij}| \quad \text{s.t.} \quad \mathbf{S} \text{ is a binary mask with transposable N:M sparsity}, \qquad (1)$$

The primary challenge in solving (1) is computational efficiency. While optimal solutions can be obtained through mixed-integer programming (e.g., Gurobi [2022]) or network flow algorithms [Hubara et al., 2021a], these approaches become computationally prohibitive for practical LLMs (e.g., LLaMA3 [Dubey et al., 2024]) where weight matrices contain billions of elements. Furthermore, as discussed in Section 4, problem (1) appears as a subproblem in each iteration of some pruning methods (e.g., [Meng et al., 2024a]), amplifying the computational demands.

We address this challenge by developing algorithms that can fully leverage GPU parallelization. Our method begins by partitioning the weight matrix into M×M blocks and reformulating problem (1) as a optimal transport problem with capacity constraints for each block. We then introduce entropy regularization and derive a solution using Dykstra's algorithm, followed by a vectorized rounding procedure that converts fractional solutions to feasible binary masks. Figure 1 illustrates our

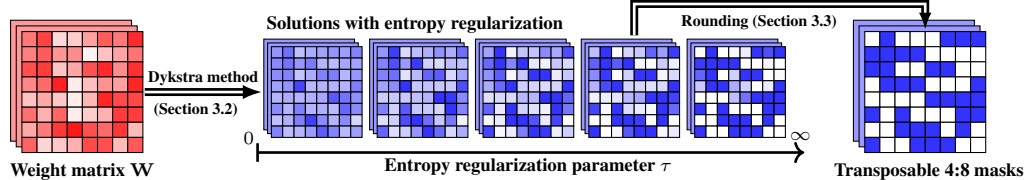

Figure 1: Efficient GPU-accelerated pipeline for transposable N:M masks generation. The weight matrix is partitioned into M×M blocks and simultaneously processed through our entropy-regularized solver and rounding procedure, leveraging tensor operations for GPU parallelization.

complete methodology. Our key innovation lies in the careful design of all algorithmic components to enable tensor-based operations, allowing simultaneous processing of millions of blocks on GPUs. This parallel implementation achieves up to two orders of magnitude speedup compared to existing methods, making our approach practical for billion-parameter language models.

## 3.1 An optimal transport reformulation

The transposable N:M sparsity constraint applies independently to each M×M submatrix of $\mathbf{S}$. Consequently, problem (1) is separable, and reduces to the following problem for each M×M block:

$$\max_{\mathbf{S} \in \{0,1\}^{M \times M}} \sum_{i,j=1}^{M} \mathbf{S}_{ij} |\mathbf{W}_{ij}| \quad \text{s.t.} \quad \sum_{i=1}^{M} \mathbf{S}_{ij} = N \, \forall j \in [M], \; \sum_{j=1}^{M} \mathbf{S}_{ij} = N \, \forall i \in [M]. \quad (2)$$

By leveraging bipartite matching polytope theory [Schrijver et al., 2003, Chapter 18], we can relax the binary variables $\mathbf{S}_{ij}$ to lie in the interval $[0, 1]$ without altering the optimal value of problem (2). Notably, any basic feasible solution of the resulting relaxed linear program corresponds to an integral optimal solution of the original problem. The relaxed problem can be expressed in matrix notation as:

$$\max_{\mathbf{S}} \langle \mathbf{S}, |\mathbf{W}| \rangle \quad \text{s.t.} \quad \mathbf{S}\mathbf{1}_M = N\mathbf{1}_M, \; \mathbf{S}^\top \mathbf{1}_M = N\mathbf{1}_M, \; \mathbf{0} \leq \mathbf{S} \leq \mathbf{1}. \quad (3)$$

We make the novel observation that problem (3) can be viewed as a capacitated optimal transport problem [Villani, 2008], where $\mathbf{S} \in [0,1]^{M \times M}$ represents the transport plan between rows and columns. Each row and column must send and receive exactly $N$ units of mass, as enforced by the constraints $\mathbf{S}\mathbf{1}_M = N\mathbf{1}_M$ and $\mathbf{S}^\top \mathbf{1}_M = N\mathbf{1}_M$. The objective is to maximize the total transported value weighted by $|\mathbf{W}|$. This connection allows us to leverage tools from optimal transport to solve the relaxed problem efficiently, as detailed in the next subsection.

## 3.2 Entropy Regularization

In practical neural network pruning scenarios, we need to efficiently solve millions of instances of problem (3) simultaneously. To enhance computational tractability, we introduce an entropy regularization term to the objective [Cuturi, 2013], reformulating the problem as:

$$\max_{\mathbf{S}} \langle \mathbf{S}, |\mathbf{W}| \rangle + \frac{1}{\tau} H(\mathbf{S}) \quad \text{s.t.} \quad \mathbf{S}\mathbf{1}_M = N\mathbf{1}_M, \; \mathbf{S}^\top \mathbf{1}_M = N\mathbf{1}_M, \; \mathbf{0} \leq \mathbf{S} \leq \mathbf{1}. \quad (4)$$

where $H(\mathbf{S}) = -\sum_{i,j=1}^{M} \mathbf{S}_{ij} \log(\mathbf{S}_{ij})$ denotes the Shannon entropy and $\tau > 0$ controls the regularization strength.

This entropy term serves two critical functions: (i) it promotes exploration by distributing "mass" more uniformly across feasible entries, preventing premature convergence to suboptimal solutions; and most importantly, (ii) it enables efficient computation via matrix-scaling algorithms that are highly parallelizable and have been GPU-accelerated in optimal transport literature [Cuturi, 2013], making them particularly suitable for our large-scale optimization requirements.

From an optimization perspective, the entropy-regularized problem can be interpreted as computing the Bregman projection of matrix $\mathbf{W}_\tau = \exp(\tau |\mathbf{W}|)$ onto the intersection of three constraint sets:

$$\mathcal{C}_1 = \{\mathbf{S} \mid \mathbf{S}\mathbf{1}_M = N\mathbf{1}_M\}, \mathcal{C}_2 = \{\mathbf{S} \mid \mathbf{S}^\top \mathbf{1}_M = N\mathbf{1}_M\}, \mathcal{C}_3 = \{\mathbf{S} \mid \mathbf{0} \leq \mathbf{S} \leq \mathbf{1}\} \quad (5)$$

with respect to the Kullback-Leibler divergence. To solve this projection problem efficiently, we employ Dykstra's algorithm [Benamou et al., 2015], which iteratively projects onto each constraint

set, as detailed in Algorithm 1 (see Appendix A.1 for detailed derivation). Critically, here each update involves only matrix-vector multiplications and element-wise operations, enabling full vectorization across millions of weight blocks and leveraging GPU acceleration.

The selection of regularization parameter $\tau$ impacts performance. A small value of $\tau$ yields solutions that poorly approximate the original problem (see Fig 1), while excessively large values impede convergence. In practice, we select $\tau$ to balance solution quality and computational efficiency. Furthermore, the entropy-regularized problem (4) only gives fractional solutions. We develop in Section 3.3 a specialized rounding procedure that can convert the approximate fractional solution into a high-quality feasible binary mask.

---

**Algorithm 1** Dykstra's algorithm for solving entropy-regularized optimal transport problem

---

**Input:** Weight matrix $\mathbf{W}$, regularization parameter $\tau > 0$, and maximum iterations $T$
**Output:** Solution $\mathbf{S}$ to problem (4)

1: Initialize $\mathbf{S}^{(0)} = \exp(\tau|\mathbf{W}|)$, and dual variable $\mathbf{Q}^{(0)} = \mathbf{1}_{M \times M}$
2: **for** $t = 0, 1, \ldots, T - 1$ **do**
3: $\quad \mathbf{S}^{(t)} \leftarrow \mathrm{Diag}\left(N/(\mathbf{S}^{(t)}\mathbf{1}_M)\right)\mathbf{S}^{(t)}$ $\qquad\qquad\qquad\qquad$ ▷ Projection onto $\mathcal{C}_1$
4: $\quad \mathbf{S}^{(t)} \leftarrow \mathbf{S}^{(t)}\mathrm{Diag}\left(N/(\mathbf{S}^{(t)\top}\mathbf{1}_M)\right)$ $\qquad\qquad\qquad$ ▷ Projection onto $\mathcal{C}_2$
5: $\quad \mathbf{S}^{(t+1)} \leftarrow \min\left(\mathbf{S}^{(t)} \odot \mathbf{Q}^{(t)}, \mathbf{1}\right)$ $\qquad\qquad\qquad\quad$ ▷ Projection onto $\mathcal{C}_3$
6: $\quad \mathbf{Q}^{(t+1)} \leftarrow \mathbf{Q}^{(t)} \odot (\mathbf{S}^{(t)} \oslash \mathbf{S}^{(t+1)})$ $\qquad\qquad\qquad$ ▷ Update dual variable
7: **return** $\mathbf{S}^{(T)}$

---

### 3.3 Sparse solution recovery via greedy selection and local search

The solution generated by Algorithm 1 is generally not binary-valued due to the entropy regularization, as illustrated in Fig.2 (1), making it unsuitable for direct use as a mask. While a simple element-wise rounding approach could be considered, it would compromise both accuracy and constraint feasibility. Therefore, we develop a novel rounding procedure that combines greedy selection with local search to obtain a high-quality feasible binary mask.

**Greedy selection:** Our approach consists of two stages. In the first greedy selection phase, we sort all elements of the approximate solution. We then iteratively assign each element to the binary mask, proceeding from largest to smallest, provided that doing so preserves the row and column sum constraints imposed by the transposable N:M sparsity. Fig.2 (2) demonstrates a binary mask obtained through this procedure under transposable 2:4 sparsity constraints.

While efficient, the greedy selection strategy can prematurely saturate certain rows or columns, preventing the placement of subsequent high-value elements and yielding suboptimal binary masks. For instance, in Fig.2 (2), the fourth row and column contain only one active element, yet the transposable N:M constraints prevent additional elements from being added.

**Local search:** To address this limitation, we introduce a novel local search procedure that refines the greedy solution. For any unsaturated row $i$ and column $j$ (i.e., containing fewer than $N$ selected elements), we explore swap-based local updates that preserve feasibility. Specifically, we explore operations that simultaneously insert two elements—one in row $i$ and one in column $j$—while removing a conflicting element to maintain the transposable N:M constraints. We enumerate all such valid insert-remove triplets and select the one that maximally increases the objective. Formally, we select candidate swap coordinates $(i', j')$ that maximize:

$$\mathrm{Swap}(i', j') := (|\mathbf{W}_{i,j'}| + |\mathbf{W}_{i',j}| - |\mathbf{W}_{i',j'}|) - \infty \cdot \left((1 - \mathbf{S}_{i',j'}) + \mathbf{S}_{i,j'} + \mathbf{S}_{i',j}\right), \quad (6)$$

where $\mathbf{S}$ denotes the current binary mask. The second term (with infinite penalty) ensures we neither insert elements already in the mask nor remove non-existent elements. When $\mathrm{Swap}(i', j') > 0$, we insert elements $(i, j')$ and $(i', j)$ while removing $(i', j')$, thereby increasing the objective value while maintaining feasibility. Fig.2 (3-4) illustrates this process, showing how adding $(2, 4)$ and $(4, 2)$ while removing $(2, 2)$ improves the objective by 0.32.

The complete rounding procedure is presented in Algorithm 2. While Hubara et al. [2021b] also proposed a greedy approach for binary mask generation, our method differs in three key aspects: (i)

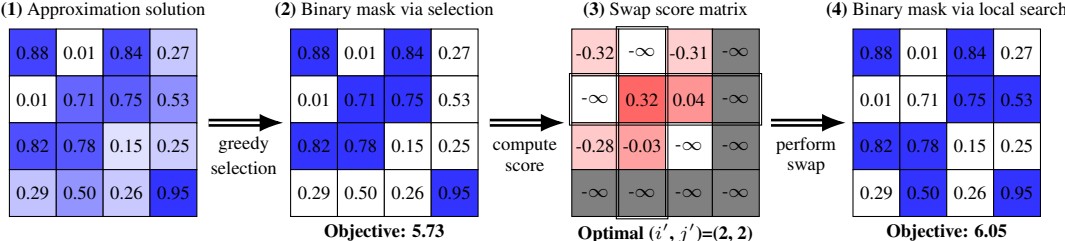

| (1) Approximation solution | (2) Binary mask via selection | (3) Swap score matrix | (4) Binary mask via local search |
| | Objective: 5.73 | Optimal $(i', j')$=(2, 2) | Objective: 6.05 |

Figure 2: Illustration of our proposed rounding procedure for generating a binary mask with transposable 2:4 sparsity. Each cell displays the absolute value of $|\mathbf{W}_{ij}|$. (1): The approximation solution obtained from Algorithm 1, the shading intensity reflects the magnitude of the approximate solution;(2): The binary mask produced through greedy selection, with row $i = 4$ and column $j = 4$ remaining unsaturated (each containing only one non-zero element); (3): Computed swap scores for candidate operations, with optimal score achieved at $(i', j') = (2, 2)$; (4): Refined binary mask after local search operation—inserting elements at positions $(4, 2)$ and $(2, 4)$ while removing $(2, 2)$.

we apply rounding to an entropy-regularized approximate solution rather than directly to the original weight matrix; (ii) our novel local search strategy can further reduce rounding error by around 50% (see Appendix B.2); and (iii) we leverage GPU to achieve significant speedup, as detailed below.

**Computational efficiency:** A key feature of our rounding approach is computational efficiency. We have fully vectorized our algorithm so that both greedy selection and local search can be performed simultaneously across all millions of blocks through tensor operations (refer to Appendix A.2 for implementation details), thereby enabling direct GPU implementation without custom CUDA kernels. This vectorization achieves up to $10^3$ times speedup compared to sequential CPU implementations (refer to Appendix B.2 for ablation studies), making our approach highly practical for large-scale network pruning applications.

---

**Algorithm 2** A binary mask generation approach based on greedy selection and local search
---
**Input:** Approximate solution $\mathbf{S}^a$ from Algorithm 1 and number of local search step $L$.
**Output:** Feasible binary solution to problem (1)
 1: Initialize $\mathbf{S} = \mathbf{0}_{M \times M}$, row counter $\mathbf{R} = \mathbf{0}_M$, column counter $\mathbf{C} = \mathbf{0}_M$
 2: $\{(i_t, j_t)\}_{t=1}^{M^2} \leftarrow \mathsf{argsort}(\{-\mathbf{S}_{ij}^a\}_{i,j=1}^M)$ $\quad\quad\quad\quad\quad$ ▷ Sort elements in descending order
 3: **for** $t = 1, 2, \ldots, M^2$ **do** $\quad\quad\quad\quad\quad\quad\quad\quad\quad\quad\quad\quad\quad$ ▷ greedy selection
 4: $\quad\quad$ **if** $R_{i_t} < N$ and $C_{j_t} < N$ **then**
 5: $\quad\quad\quad$ Set $\mathbf{S}_{i_t, j_t} \leftarrow 1$
 6: $\quad\quad\quad$ Update $(\mathbf{R}_{i_t}, \mathbf{C}_{j_t}) \leftarrow (\mathbf{R}_{i_t} + 1, \mathbf{C}_{j_t} + 1)$

 7: **for** $t = 1, 2, \ldots, L$ **do** $\quad\quad\quad\quad\quad\quad\quad\quad\quad\quad\quad\quad\quad$ ▷ $L$ local search steps
 8: $\quad\quad$ **if** $R_i = N$, $C_i = N$, $\forall i \in [M]$ **then**
 9: $\quad\quad\quad$ **break**
10: $\quad\quad$ Pick $i, j \in [M]$ such that $\mathbf{R}_i < N$, $\mathbf{C}_j < N$.
11: $\quad\quad$ Select $(i', j') = \arg\max_{i', j'} \mathrm{Swap}(i', j')$ as defined in Eq.(6).
12: $\quad\quad$ **if** $\mathrm{Swap}(i', j') > 0$ **then**
13: $\quad\quad\quad$ Set $(\mathbf{S}_{i', j'}, \mathbf{S}_{i', j}, \mathbf{S}_{i, j'}) \leftarrow (0, 1, 1)$, $(\mathbf{R}_i, \mathbf{C}_j) \leftarrow (\mathbf{R}_i + 1, \mathbf{C}_j + 1)$.
14: **return** Feasible binary mask $\mathbf{S}$

---

## 4 Layer-wise reconstruction with transposable N:M sparsity

We integrate *TSENOR* within leading LLM layer-wise pruning approaches to obtain transposable N:M sparse networks. These masks enable *efficient fine-tuning* with acceleration in both forward and backward passes. Note that layer-wise pruning minimizes the discrepancy in outputs between dense and pruned layers, formulated as

$$\min_{\mathbf{W}} \ (1/2)\|\mathbf{X}(\mathbf{W} - \widehat{\mathbf{W}})\|_F^2 + (\lambda/2)\|\mathbf{W} - \widehat{\mathbf{W}}\|_F^2 \ \text{s.t.} \ \mathbf{W} \in \mathcal{T}, \quad\quad\quad (7)$$

where $\widehat{\mathbf{W}}$ is the pre-trained weights, $\mathbf{X}$ represents input activations, and $\mathcal{T}$ denotes set of matrices with transposable N:M sparsity. We demonstrate integration with Wanda [Sun et al., 2023], SparseGPT [Frantar and Alistarh, 2023], and ALPS [Meng et al., 2024a].

**Integration with Wanda:** Wanda evaluates weight importance using the product of weight magnitude and corresponding input feature norm, performing magnitude pruning based on this importance score. Our integration with Wanda involves solving problem (1) with $\mathbf{W}$ replaced by $\mathbf{W}'$ with entries $\mathbf{W}'_{ij} = \mathbf{W}_{ij}\|\mathbf{X}_{:,i}\|_2$ to get the pruning mask and setting all elements outside the mask to zero.

**Integration with SparseGPT:** SparseGPT traverses $\mathbf{W}$ left-to-right in groups of M columns, pruning each group $\mathbf{W}^G$ according to OBS scores [Hassibi and Stork, 1992], and updating $\mathbf{W}^G$ and remaining columns accordingly. To achieve transposable N:M sparsity, we substitute the pruning step with *TSENOR*, solving problem (1) with $\mathbf{W}$ replaced by the matrix with entries $(-\mathbf{W}^G_{ij}/[\mathbf{H}^{-1}]_{jj})$—please refer to the SparseGPT paper for algorithmic details and efficient updates of $\mathbf{H}^{-1}$.

**Integration with ALPS:** Integrating *TSENOR* with ALPS poses technical challenges as ALPS' original update rules and convergence guarantees do not directly apply to our setting. We derive modified updates and establish new convergence guarantees as below. Following ALPS framework, we addresses layer-wise pruning problem through the ADMM approach [Boyd et al., 2011]. We introduce an auxiliary variable $\mathbf{D}$ that replicates $\mathbf{W}$ and consider the following augmented Lagrangian:

$$L_\rho(\mathbf{W}, \mathbf{D}, \mathbf{V}) = \frac{1}{2}\|\mathbf{X}(\mathbf{W} - \widehat{\mathbf{W}})\|_F^2 + \frac{\lambda}{2}\|\mathbf{W} - \widehat{\mathbf{W}}\|_F^2 + \mathbb{I}_\mathcal{T}(\mathbf{D}) + \langle \mathbf{V}, \mathbf{W} - \mathbf{D}\rangle + \frac{\rho}{2}\|\mathbf{W} - \mathbf{D}\|_F^2, \quad (8)$$

where $\mathbb{I}_\mathcal{T}(\mathbf{D})$ is the indicator function that equals zero when $\mathbf{D} \in \mathcal{T}$ and infinity otherwise and $\rho > 0$ is the penalty parameter. ADMM minimizes this augmented Lagrangian by alternately updating $\mathbf{W}$ and $\mathbf{D}$, followed by a dual update for $\mathbf{V}$.

**Proposition 1.** *At iteration $t$, the ADMM update rules are given by:*

$$\mathbf{W}^{(t+1)} = \arg\min_\mathbf{W} L_\rho(\mathbf{W}, \mathbf{D}^{(t)}, \mathbf{V}^{(t)}) = (\mathbf{H} + \rho\mathbf{I})^{-1}(\mathbf{H}\widehat{\mathbf{W}} - \mathbf{V}^{(t)} + \rho\mathbf{D}^{(t)}),$$
$$\mathbf{D}^{(t+1)} = \arg\min_\mathbf{D} L_\rho(\mathbf{W}^{(t+1)}, \mathbf{D}, \mathbf{V}^{(t)}) = (\mathbf{W}^{(t+1)} + \mathbf{V}^{(t)}/\rho) \odot \mathbf{S}^{(t+1)}, \quad (9)$$
$$\mathbf{V}^{(t+1)} = \mathbf{V}^{(t)} + \rho(\mathbf{W}^{(t+1)} - \mathbf{D}^{(t+1)}),$$

*where $\mathbf{H} = \mathbf{X}^\top\mathbf{X} + \lambda\mathbf{I}$. Above, $\mathbf{S}^{(t+1)}$ is the solution to the following problem:*

$$\max_\mathbf{S} \sum_{i,j} \mathbf{S}_{ij}(\mathbf{W}^{(t+1)}_{ij} + \mathbf{V}^{(t)}_{ij}/\rho)^2 \ \text{s.t. } \mathbf{S} \text{ is a binary mask with transposable N:M sparsity} \quad (10)$$

*with the same structure as (1), and we can directly apply our binary mask solver from Section 3.*

The key distinction between our modified ALPS framework and the original is in the $\mathbf{D}$-update step. While standard ALPS employs direct projection for unstructured or non-transposable N:M sparsity, our approach solves the transposable N:M sparsity constraint using our proposed entropy-regularized algorithm followed by a rounding procedure. Our approach may not find the globally optimal mask, but we can still establish theoretical convergence guarantees as in Theorem 1. Complete statements and proofs of Proposition 1 and Theorem 1 are provided in Appendix A.1.

**Theorem 1.** *(Informal) Under mild assumptions on penalty parameter $\rho$ and sufficient accuracy of each $\mathbf{D}$-update, there exists a matrix $\bar{\mathbf{W}}$ such that $\mathbf{D}^{(t)} \to \bar{\mathbf{W}}$ and $\mathbf{W}^{(t)} \to \bar{\mathbf{W}}$ as $t \to \infty$.*

# 5 Experiments

This section evaluates the efficiency and effectiveness of our proposed transposable binary mask solver and demonstrates its performance when integrated into existing N:M pruning frameworks. Detailed experimental setup and reproducibility information are provided in Appendix B.1, with ablation studies and additional experimental results presented in Appendix B.2.

## 5.1 Performance on a single matrix

We evaluate our proposed transposable binary mask solver against several approaches: (i) Network Flow method [Hubara et al., 2021b], which guarantees optimal solutions through bipartite matching algorithms; (ii) cuPDLP [Lu and Yang, 2023], a general-purpose GPU-accelerated linear programming

solver; (iii) 2-Approximation [Hubara et al., 2021b], a greedy-based heuristic; (iv) Bi-NM adopted from [Zhang et al., 2023], which sequentially applies row-wise and then column-wise N:M sparsity; and (v) Max1000, a randomized baseline that generates 1000 feasible masks and selects the best one. Our experiments focus on transposable N:M sparsity with M≥8, as smaller patterns (M=4) can already be optimally and efficiently solved [Hu et al., 2024].

We first examine solution quality of each approach through relative error against optimal solutions, excluding Network Flow and cuPDLP which guarantee optimality. We evaluate two variants of our method (1) Entropy: Algorithm 1 with simple row-then-column N:M sparsity rounding, and (2) *TSENOR*: our full pipeline combining entropy regularization with specialized rounding (Algorithm 2). As shown in Figure 3, *TSENOR* achieves remarkably low relative error (1-10%) compared to 2-Approximation, the best competing heuristic. The specialized rounding procedure alone also contributes substantially, reducing error by up to 10× compared to simple rounding (i.e., Entropy), validating the effectiveness of both components in our pipeline. In contrast, simpler approaches like Bi-NM and Max1000 exhibit substantially higher relative errors (up to 50%), highlighting the inherent difficulty of the transposable N:M sparsity problem and the value of our advanced solver.

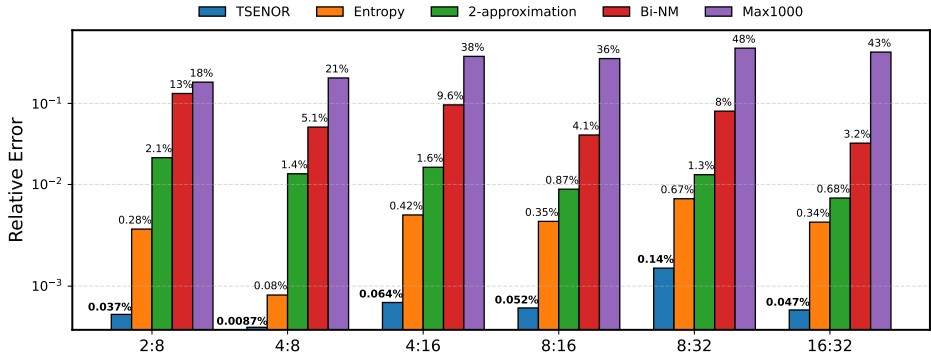

Figure 3: Solution quality comparison for transposable N:M mask generation. For various N:M sparsity patterns, we evaluate all methods on 100 M×M blocks sampled from LLaMA3 [Dubey et al., 2024] model weights. We report the average relative error, defined as $(f(\mathbf{S}^*) - f(\mathbf{S}))/f(\mathbf{S}^*)$, where $\mathbf{S}^*$ is the optimal support and $f$ is the objective defined in (1).

Next, we benchmark computational efficiency on matrices of increasing size. We compare our method with two CPU-implemented approaches (Network Flow and 2-Approximation) and one GPU-accelerated method (cuPDLP) across various hardware platforms. Table 1 shows our method consistently achieves the fastest runtimes, delivering up to 300× speedup compared to methods guaranteeing optimal solutions. While Bi-NM and Max1000 (not shown in the table) offer negligible runtime overhead, their poor solution quality renders them impractical. This implies that our solver provides the optimal balance between solution quality and efficiency for large-scale network pruning.

| Matrix Size | Network Flow | 2-Approximation | cuPDLP | | | *TSENOR* | | |
|---|---|---|---|---|---|---|---|---|
| | | | V100 | A100 | H100 | V100 | A100 | H100 |
| 512 × 512 | 1.82 (±0.12) | 0.13 (±0.01) | 14.6 (±0.51) | 12.1 (±0.66) | 7.82 (±0.56) | 0.11 (±0.00) | 0.16 (±0.00) | **0.08** (±0.00) |
| 2048 × 2048 | 23.3 (±0.89) | 0.35 (±0.01) | 20.9 (±0.82) | 18.0 (±1.01) | 11.9 (±1.08) | 0.27 (±0.00) | 0.20 (±0.00) | **0.12** (±0.00) |
| 8192 × 8192 | 350 (±5.22) | 3.23 (±0.09) | 252 (±9.65) | 28.1 (±1.04) | 16.8 (±0.95) | 3.26 (±0.00) | 1.74 (±0.00) | **1.06** (±0.00) |

Table 1: Runtime (seconds) for transposable 8:16 sparsity. For GPU methods, we test on NVIDIA V100-PCIe-32GB, A100-PCIe-40GB, and H100-PCIe-80GB. CPU methods use 16-core parallel processing. Results are averaged over 10 trials with standard deviations in parentheses.

## 5.2 LLaMA with transposable N:M sparsity

We evaluate our transposable N:M solver integrated into existing pruning approaches: Wanda [Sun et al., 2023], SparseGPT [Frantar and Alistarh, 2023] and ALPS [Meng et al., 2024a], using LLaMA-3.2 [Dubey et al., 2024] models with 1 to 8 billion parameters. Performance is assessed via perplexity

and zero-shot benchmarks. Perplexity is computed following HuggingFace's methodology [Per, 2022] with full stride on raw-WikiText2 [Merity et al., 2017], PTB [Marcus et al., 1994], and C4 [Raffel et al., 2020] validation subset. Zero-shot evaluation uses LM Harness [Gao et al.] on PIQA [Bisk et al., 2020], ARC-E and ARC-C [Clark et al., 2018], Hellaswag [Zellers et al., 2019], Winogrande [Sakaguchi et al., 2021], RTE [Poliak, 2020], OpenbookQA [Banerjee et al., 2019], and BoolQ [Clark et al., 2019].

### 5.2.1 Trade-offs: benefits and costs of transposable N:M sparsity

Transposable N:M sparsity enables backward pass acceleration beyond standard N:M sparsity, but imposes stricter constraints that potentially impact model performance. We quantify this trade-off through experiments. Fig. 4 (Upper) compares perplexity of LLaMA3.2-8B when pruned to different sparsity patterns, while Fig. 4 (Lower) illustrates computational speedup versus dense computation. Our results demonstrate that the performance gap between transposable N:M and standard N:M diminishes dramatically as M increases from 4 to 32 (by approximately 90%), while transposable N:M consistently delivers superior speedup (3.3× at 75% sparsity) regardless of M value. This indicates that transposable N:M sparsity with larger M values offers an excellent practical trade-off, highlighting the importance of our efficient solver for arbitrary transposable sparsity patterns.

### 5.2.2 Integration with various pruning frameworks

We evaluate our solver's effectiveness when integrated into different pruning methods. Table 2 presents the performance of LLaMA3.2-3B after one-shot pruning with various frameworks. As expected, applying transposable N:M sparsity directly with Wanda or SparseGPT results in significant accuracy degradation. However, integrating our solver with ALPS, which minimizes layerwise reconstruction error, successfully recovers most of the performance loss. Notably, our approach combined with ALPS produces transposable N:M sparse models that slightly outperform standard N:M sparse models obtained through SparseGPT, while providing additional computational benefits.

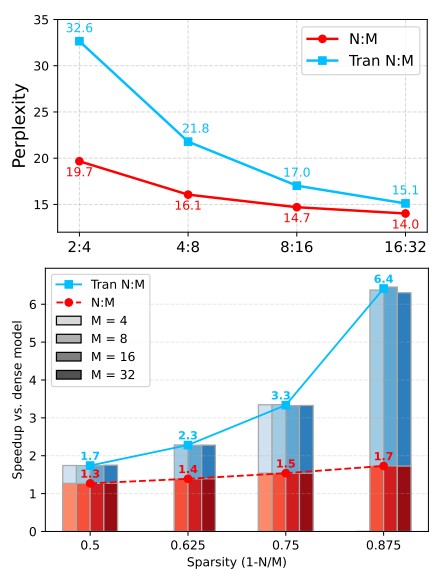

Figure 4: Upper: perplexity of LLaMA3.2-8B pruned by ALPS with *TSENOR*, comparing standard and transposable N:M sparsity across different N:M patterns. (2) Lower: Computational speedup for forward and backward matrix operations in LLaMA3.2-8B, comparing (transposable) N:M sparse matrices (via nmSPMM [Ma et al., 2025]) against dense matrices (via NVIDIA cuBLAS).

| N:M Sparsity | Algorithm | Transpose | Perplexity (↓) | | | Zero-shot (↑) | | | | | | | | |
|---|---|---|---|---|---|---|---|---|---|---|---|---|---|---|
| | | | C4 | WT2 | PTB | PIQA | HS | ARC-E | ARC-C | WG | RTE | OQA | BoolQ | Avg |
| 8:32 | SparseGPT | ✗ | 139.67 | 120.22 | 183.01 | 53.86 | 28.80 | 29.50 | 21.67 | 48.46 | 54.15 | 27.20 | 62.23 | 40.74 |
| | ALPS | ✗ | 80.12 | 82.28 | 110.73 | 56.64 | 31.27 | 32.66 | 19.80 | 51.38 | 52.35 | 27.00 | 61.99 | 41.63 |
| | TSENOR+Wanda | ✓ | 73379.13 | 100992.48 | 289678.69 | 50.98 | 26.06 | 24.83 | 27.99 | 50.12 | 52.71 | 26.60 | 55.26 | 39.32 |
| | TSENOR+SparseGPT | ✓ | 239.08 | 302.73 | 378.69 | 52.56 | 27.33 | 28.62 | 23.04 | 49.80 | 52.71 | 26.40 | 41.35 | 37.72 |
| | TSENOR+ALPS | ✓ | 111.36 | 163.18 | 178.60 | 54.90 | 28.92 | 30.43 | 20.39 | 50.99 | 53.07 | 26.60 | 61.22 | 40.81 |
| 16:32 | SparseGPT | ✗ | 18.11 | 13.02 | 20.37 | 73.12 | 62.27 | 62.67 | 34.81 | 65.75 | 60.65 | 35.20 | 71.53 | 58.25 |
| | ALPS | ✗ | 16.74 | 12.06 | 18.78 | 73.56 | 64.10 | 64.48 | 36.52 | 66.54 | 57.40 | 39.00 | 72.81 | 59.30 |
| | TSENOR+Wanda | ✓ | 436.13 | 262.78 | 363.91 | 57.29 | 30.54 | 37.79 | 21.25 | 50.28 | 52.71 | 25.60 | 49.82 | 40.66 |
| | TSENOR+SparseGPT | ✓ | 20.29 | 14.80 | 23.81 | 72.36 | 58.69 | 59.64 | 33.02 | 63.06 | 53.79 | 34.20 | 65.99 | 55.10 |
| | TSENOR+ALPS | ✓ | 18.03 | 13.08 | 20.43 | 72.96 | 61.43 | 63.22 | 38.05 | 63.46 | 57.40 | 35.60 | 72.32 | 58.06 |

Table 2: Performance analysis for (transposable) N:M pruning on LLaMA3.2-3B model. Lower values are preferred for perplexity, and higher values are preferred for zero-shot tasks.

### 5.2.3 Fine-tuning transposable N:M sparse models

We fine-tune transposable N:M sparse models pruned with our approach (`TSENOR+ALPS`) on C4 and compare their perplexity against Bi-NM [Zhang et al., 2023], which trains a non-transposable N:M network using gradients approximately calculated through a transposable mask. As shown in Figure 5, Bi-NM performs slightly better when M=4, but our approach achieves progressively lower perplexity as M increases. This is because transposable sparsity with larger M values has smaller impact on model performance, and our method's use of exact gradients during fine-tuning which leading to more effective parameter update. Refer to Appendix B.2.6 for a comprehensive comparison between `TSENOR+ALPS` and Bi-NM.

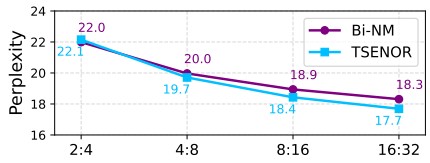

Figure 5: Perplexity comparison of LLaMA3.2-1B under two pruning approaches: (1) `TSENOR+ALPS` followed by fine-tuning, and (2) standard N:M pruning followed by retraining via Bi-NM.

## 6 Limitations and future work

While our work presents efficient algorithms for computing transposable N:M masks and transposable N:M sparse models and demonstrates computational speedup on matrix operations, several directions remain unexplored. Developing an end-to-end training pipeline that fully realizes the practical acceleration from transposable N:M sparsity would strengthen the applicability of our method. Additionally, comparing the computational cost and model quality trade-offs between transposable N:M sparsity and parameter-efficient fine-tuning methods (e.g., LoRA [Hu et al., 2021], SPP [Lu et al., 2024]) would provide valuable guidance for practitioners. Another promising direction is incorporating channel-wise permutations of weight matrices before computing transposable N:M masks. While this approach has proven successful for standard N:M sparsity [Pool and Yu, 2021], extending it to the transposable setting remains unexplored. Despite introducing a complex combinatorial optimization problem, such extension might substantially improve pruning quality. Similarly, it would be interesting to explore loss functions that go beyond simple layerwise reconstruction error [Lucas and Mazumder]. We believe these directions represent important future paths for advancing practical sparse model deployment.

## 7 Conclusion

In this paper we present *TSENOR*—a novel efficient algorithm for generating high-quality transposable N:M sparse masks. *TSENOR* solves an entropy-regularized optimal transport problem and applies a new rounding procedure combining greedy selection with local search. By using tensor operations throughout, our method achieves significant speedup through GPU acceleration and scales to billion-parameter language models. We can incorporate *TSENOR* within several existing layerwise pruning frameworks to create transposable N:M sparse LLMs with any N:M pattern. Our experiments show that networks with larger M values (e.g., 16:32) provide the same computational benefits as smaller M values but with much less impact on model accuracy, demonstrating our method's practical value.

## 8 Acknowledgements

This research is supported in part by grants from the Office of Naval Research (N000142512504, N000142212665). We acknowledge the MIT Engaging cluster for providing HPC resources that have contributed to the research results reported within this paper. Additionally, we thank Google for providing us with Google Cloud Credits. We thank Shibal Ibrahim, Ryan Lucas, and Gabriel Afriat for their helpful discussions.

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

# A  Technique Details

## A.1  Proofs of main results

### A.1.1  Derivation of Dykstra's algorithm

In Section 3.2, we focus on solving the capacity-constrained optimal transport problem with entropy regularization:

$$\max_{\mathbf{S}} \langle \mathbf{S}, |\mathbf{W}| \rangle + \frac{1}{\tau} H(\mathbf{S}) \text{ s.t. } \mathbf{S}\mathbf{1}_M = N\mathbf{1}_M, \ \mathbf{S}^\top \mathbf{1}_M = N\mathbf{1}_M, \ \mathbf{0} \leq \mathbf{S} \leq \mathbf{1}. \quad (11)$$

The objective of (11) can be reformulated as follows:

$$
\begin{aligned}
\langle \mathbf{S}, |\mathbf{W}| \rangle + \frac{1}{\tau} H(\mathbf{S}) &= \sum_{i,j=1}^{M} \mathbf{S}_{ij} |\mathbf{W}_{ij}| - \frac{1}{\tau} \mathbf{S}_{ij} \log(\mathbf{S}_{ij}) \\
&= -\frac{1}{\tau} \sum_{i,j=1}^{M} \mathbf{S}_{ij} \log \left( \frac{\mathbf{S}_{ij}}{e^{\tau |\mathbf{W}_{ij}|}} \right) \\
&= -\frac{1}{\tau} D_{\mathrm{KL}}(\mathbf{S} \, || \, \mathbf{W}_\tau)
\end{aligned}
\quad (12)
$$

where $\mathbf{W}_\tau = \exp(\tau |\mathbf{W}|)$ and $D_{\mathrm{KL}}(\mathbf{S} \, || \, \mathbf{W}_\tau)$ denotes the Kullback-Leibler divergence of $\mathbf{S}$ from $\mathbf{W}_\tau$. Therefore, the optimization problem can be reformulated as:

$$\min_{\mathbf{S}} D_{\mathrm{KL}}(\mathbf{S} \, || \, \mathbf{W}_\tau) \text{ s.t. } \mathbf{S} \in \bigcap_{i=1}^{3} \mathcal{C}_i, \quad (13)$$

where

$$\mathcal{C}_1 = \{\mathbf{S} \mid \mathbf{S}\mathbf{1}_M = N\mathbf{1}_M\}, \ \mathcal{C}_2 = \{\mathbf{S} \mid \mathbf{S}^\top \mathbf{1}_M = N\mathbf{1}_M\}, \ \mathcal{C}_3 = \{\mathbf{S} \mid \mathbf{0} \leq \mathbf{S} \leq \mathbf{1}\}. \quad (14)$$

This formulation can be interpreted from the perspective of Bregman projections [Bregman, 1967]. The Kullback-Leibler divergence is a special case of the Bregman divergence generated by the negative entropy function. For any Bregman divergence $D_\phi$, the Bregman projection of a point $\mathbf{y}$ onto a convex set $\mathcal{C}$ is defined as:

$$\mathcal{P}_{\mathcal{C}}^{\phi}(\mathbf{y}) = \arg\min_{\mathbf{x} \in \mathcal{C}} D_\phi(\mathbf{x}, \mathbf{y}). \quad (15)$$

In our context, problem (13) represents the Bregman projection of $\mathbf{W}_\tau$ onto the intersection of the constraint sets $\bigcap_{i=1}^{3} \mathcal{C}_i$ with respect to the KL divergence. While computing this projection directly is challenging, Dykstra's algorithm [Dykstra, 1983] provides an iterative approach by alternately projecting onto each constraint set.

Dykstra's algorithm for Bregman projections begins by initializing:

$$\mathbf{S}^{(0)} = \mathbf{W}_\tau \quad \text{and} \quad \mathbf{Q}_1^{(0)} = \mathbf{Q}_2^{(0)} = \mathbf{Q}_3^{(0)} = \mathbf{1}_{M \times M}. \quad (16)$$

At each iteration $t$, for each constraint set $\mathcal{C}_i$ ($i = 1, 2, 3$), the algorithm performs:

$$\mathbf{S}^{(t+i/3)} = \mathcal{P}_{\mathcal{C}_i}^{\mathrm{KL}}(\mathbf{S}^{(t+(i-1)/3)} \odot \mathbf{Q}_i^{(t)}), \quad \text{and} \quad \mathbf{Q}_i^{(t+1)} = \mathbf{Q}_i^{(t)} \odot (\mathbf{S}^{(t+(i-1)/3)} \oslash \mathbf{S}^{(t+i/3)}) \quad (17)$$

where $\mathcal{P}_{\mathcal{C}}^{\mathrm{KL}}(\mathbf{S}) = \arg\min_{\gamma \in \mathcal{C}} \mathrm{KL}(\gamma \, || \, \mathbf{S})$ denotes the KL projection onto set $\mathcal{C}$, $\odot$ and $\oslash$ denote element-wise multiplication and division, respectively. For our specific constraint sets, the projection operations can be derived analytically:

1. For the row sum constraint $\mathcal{C}_1$, the KL projection has the closed-form solution:

$$\mathcal{P}_{\mathcal{C}_1}^{\mathrm{KL}}(\mathbf{S}) = \mathrm{Diag}\left( \frac{N}{\mathbf{S}\mathbf{1}_M} \right) \mathbf{S} \quad (18)$$

This is a row-wise scaling operation that ensures each row sums to $N$.

2. Similarly, for the column sum constraint $\mathcal{C}_2$, the projection is:

$$\mathcal{P}_{\mathcal{C}_2}^{\mathrm{KL}}(\mathbf{S}) = \mathbf{S}\mathrm{Diag}\left(\frac{N}{\mathbf{S}^\top \mathbf{1}_M}\right) \tag{19}$$

This performs column-wise scaling to satisfy the column sum constraint.

3. For the capacity constraint $\mathcal{C}_3$, the projection is:

$$\mathcal{P}_{\mathcal{C}_3}^{\mathrm{KL}}(\mathbf{S}) = \min(\mathbf{S}, \mathbf{1}) \tag{20}$$

This element-wise thresholding operation ensures all entries remain bounded by 1.

Notably, for constraints $\mathcal{C}_1$ and $\mathcal{C}_2$, the dual variables $\mathbf{Q}_1^{(t)}$ and $\mathbf{Q}_2^{(t)}$ can be eliminated from the implementation. This simplification arises because the projection onto $\mathcal{C}_1$ scales each row by a constant factor to ensure the row sum equals $N$. When we subsequently update $\mathbf{Q}_1^{(t)}$ according to the ratio between the pre-projection and post-projection matrices, these scaling factors are precisely encoded in $\mathbf{Q}_1^{(t)}$. However, in the next iteration, when we project $\mathbf{S}^{(t)} \odot \mathbf{Q}_1^{(t)}$ onto $\mathcal{C}_1$, the row-wise scaling operation will normalize each row to sum to $N$ regardless of the initial scaling. Thus, incorporating $\mathbf{Q}_1^{(t)}$ has no effect on the final projection result. An identical argument applies to the column constraint $\mathcal{C}_2$ and its corresponding dual variable $\mathbf{Q}_2^{(t)}$. As a result, we only need to maintain a single dual variable $\mathbf{Q}$ corresponding to the capacity constraint $\mathcal{C}_3$. The complete simplified algorithm is presented in Algorithm 1.

### A.1.2 Proof of Proposition 1

We derive the ADMM update rules by sequentially minimizing the augmented Lagrangian with respect to each variable. First, we consider the $\mathbf{W}$-update: $\mathbf{W}^{(t+1)} = \arg\min_{\mathbf{W}} L_\rho(\mathbf{W}, \mathbf{D}^{(t)}, \mathbf{V}^{(t)})$. Since $L_\rho(\mathbf{W}, \mathbf{D}^{(t)}, \mathbf{V}^{(t)})$ is a quadratic function of $\mathbf{W}$, we can find the minimizer by setting the gradient to zero:

$$\nabla_{\mathbf{W}} L_\rho(\mathbf{W}, \mathbf{D}^{(t)}, \mathbf{V}^{(t)}) = (\mathbf{X}^\top \mathbf{X} + \lambda \mathbf{I})(\mathbf{W} - \widehat{\mathbf{W}}) + \mathbf{V}^{(t)} + \rho(\mathbf{W} - \mathbf{D}^{(t)}) = \mathbf{0} \tag{21}$$

Letting $\mathbf{H} = \mathbf{X}^\top \mathbf{X} + \lambda \mathbf{I}$, this yields the closed-form solution:

$$\mathbf{W}^{(t+1)} = (\mathbf{H} + \rho\mathbf{I})^{-1}(\mathbf{H}\widehat{\mathbf{W}} - \mathbf{V}^{(t)} + \rho\mathbf{D}^{(t)}) \tag{22}$$

Next, we consider the $\mathbf{D}$-update: $\mathbf{D}^{(t+1)} = \arg\min_{\mathbf{D}} L_\rho(\mathbf{W}^{(t+1)}, \mathbf{D}, \mathbf{V}^{(t)})$. This is equivalent to solving:

$$\min_{\mathbf{D} \in \mathcal{T}} \langle \mathbf{V}^{(t)}, \mathbf{W}^{(t+1)} - \mathbf{D}\rangle + \frac{\rho}{2}\|\mathbf{W}^{(t+1)} - \mathbf{D}\|_F^2 \tag{23}$$

Completing the square and removing constant terms with respect to $\mathbf{D}$, we have:

$$\min_{\mathbf{D} \in \mathcal{T}} \frac{\rho}{2}\left\|\mathbf{D} - \left(\mathbf{W}^{(t+1)} + \mathbf{V}^{(t)}/\rho\right)\right\|_F^2 \tag{24}$$

The constraint $\mathbf{D} \in \mathcal{T}$ requires $\mathbf{D}$ to have a specific sparsity pattern, which we can represent using a binary mask $\mathbf{S}$ where $\mathbf{S}_{ij} = 1$ indicates that $\mathbf{D}_{ij}$ can be non-zero. Given a fixed support defined by $\mathbf{S}$, the optimal values for non-zero elements are exactly those of the unconstrained solution $\mathbf{W}^{(t+1)} + \mathbf{V}^{(t)}/\rho$. Thus, we can express $\mathbf{D}$ as:

$$\mathbf{D} = \left(\mathbf{W}^{(t+1)} + \mathbf{V}^{(t)}/\rho\right) \odot \mathbf{S} \tag{25}$$

Our objective now becomes determining the optimal binary mask $\mathbf{S}$. Substituting this representation of $\mathbf{D}$ into the objective:

$$\frac{\rho}{2}\left\|(\mathbf{W}^{(t+1)} + \mathbf{V}^{(t)}/\rho) \odot \mathbf{S} - (\mathbf{W}^{(t+1)} + \mathbf{V}^{(t)}/\rho)\right\|_F^2 = \frac{\rho}{2}\sum_{i,j}\left[(\mathbf{W}_{ij}^{(t+1)} + \mathbf{V}_{ij}^{(t)}/\rho)(1 - \mathbf{S}_{ij})\right]^2$$

$$= \frac{\rho}{2}\sum_{i,j}(\mathbf{W}_{ij}^{(t+1)} + \mathbf{V}_{ij}^{(t)}/\rho)^2(1 - \mathbf{S}_{ij}) \tag{26}$$

The last equality follows since $\mathbf{S}_{ij} \in \{0, 1\}$ implies $(1 - \mathbf{S}_{ij})^2 = (1 - \mathbf{S}_{ij})$. Minimizing this expression is equivalent to maximizing:

$$\max_{\mathbf{S}} \sum_{i,j} \mathbf{S}_{ij} \left( \mathbf{W}_{ij}^{(t+1)} + \mathbf{V}_{ij}^{(t)}/\rho \right)^2 \quad \text{s.t.} \quad \mathbf{S} \text{ is a binary mask with transposable N:M sparsity}$$

(27)

Once the optimal mask $\mathbf{S}$ is determined, we recover $\mathbf{D}^{(t+1)}$ as:

$$\mathbf{D}^{(t+1)} = \left( \mathbf{W}^{(t+1)} + \mathbf{V}^{(t)}/\rho \right) \odot \mathbf{S}$$

(28)

Finally, the dual update follows the standard ADMM methodology:

$$\mathbf{V}^{(t+1)} = \mathbf{V}^{(t)} + \rho(\mathbf{W}^{(t+1)} - \mathbf{D}^{(t+1)})$$

(29)

In practice, we employ an adaptive penalty parameter $\rho_t$ that varies across iterations, resulting in the following update rules:

$$\begin{aligned} \mathbf{W}^{(t+1)} &= (\mathbf{H} + \rho_t \mathbf{I})^{-1}(\mathbf{H}\widehat{\mathbf{W}} - \mathbf{V}^{(t)} + \rho_t \mathbf{D}^{(t)}), \\ \mathbf{D}^{(t+1)} &= \left( \mathbf{W}^{(t+1)} + \mathbf{V}^{(t)}/\rho_t \right) \odot \mathbf{S}^{(t+1)}, \\ \mathbf{V}^{(t+1)} &= \mathbf{V}^{(t)} + \rho_t(\mathbf{W}^{(t+1)} - \mathbf{D}^{(t+1)}) \end{aligned}$$

(30)

This completes the proof of Proposition 1.

### A.1.3 Convergence of update (30)

We begin by formalizing our assumptions regarding the penalty parameter sequence and the quality of each $\mathbf{D}^{(t+1)}$ update.

**Assumption 1.** *The penalty parameters $\{\rho_t\}_{t=1}^{\infty}$ in (30) are chosen to be an increasing sequence such that $\sum_{t=1}^{\infty} 1/\rho_t$ converges. Additionally, the binary mask $\mathbf{S}^{(t+1)}$ obtained from our solver does not decrease the objective in (10) compared to $\mathbf{S}^{(t)}$, i.e.,*

$$\sum_{i,j} \mathbf{S}_{ij}^{(t+1)} \left( \mathbf{W}_{ij}^{(t+1)} + \mathbf{V}_{ij}^{(t)}/\rho_t \right)^2 \geq \sum_{i,j} \mathbf{S}_{ij}^{(t)} \left( \mathbf{W}_{ij}^{(t+1)} + \mathbf{V}_{ij}^{(t)}/\rho_t \right)^2.$$

(31)

Assumption 1 is mild in practice. To ensure $\sum_{t=1}^{\infty} 1/\rho_t$ converges, we can simply select $\{\rho_t\}_{t=1}^{\infty}$ as an increasing geometric sequence. The second condition is readily satisfied by comparing the objective value of $\mathbf{S}^{(t+1)}$ obtained from our binary mask solver with that of $\mathbf{S}^{(t)}$, and defaulting to $\mathbf{S}^{(t+1)} = \mathbf{S}^{(t)}$ whenever the new solution would decrease the objective (though empirically, this safeguard never triggers).

We now restate the convergence theorem in its complete form:

**Theorem 1.** *Under Assumption 1, let $\{\mathbf{D}^{(t)}\}_{t=0}^{\infty}$ and $\{\mathbf{W}^{(t)}\}_{t=0}^{\infty}$ be the sequences generated according to (30). Then there exists a matrix $\bar{\mathbf{W}}$ such that $\mathbf{D}^{(t)} \to \bar{\mathbf{W}}$ and $\mathbf{W}^{(t)} \to \bar{\mathbf{W}}$ as $t \to \infty$.*

**Proof of Theorem 1** The majority of this proof follows from the convergence analysis in [Meng et al., 2024a, Theorem 1], as we employ identical $\mathbf{W}$ and $\mathbf{V}$ update rules. The critical distinction lies in our $\mathbf{D}$-update step, specifically in establishing the following inequality, which cannot be directly borrowed from the original proof:

$$\left\| \mathbf{D}^{(t+1)} - (\mathbf{W}^{(t+1)} + \mathbf{V}^{(t)}/\rho_t) \right\|_F^2 \leq \left\| \mathbf{D}^{(t)} - (\mathbf{W}^{(t+1)} + \mathbf{V}^{(t)}/\rho_t) \right\|_F^2.$$

(32)

To establish this inequality, we proceed as follows:

$$
\begin{aligned}
\left\| \mathbf{D}^{(t+1)} - \left( \mathbf{W}^{(t+1)} + \mathbf{V}^{(t)}/\rho_t \right) \right\|_F^2 &= \left\| \left( \mathbf{W}^{(t+1)} + \mathbf{V}^{(t)}/\rho_t \right) \odot (1 - \mathbf{S}^{(t+1)}) \right\|_F^2 \\
&= \sum_{i,j} (1 - \mathbf{S}_{ij}^{(t+1)}) \left( \mathbf{W}_{ij}^{(t+1)} + \mathbf{V}_{ij}^{(t)}/\rho_t \right)^2 \\
&\leq \sum_{i,j} (1 - \mathbf{S}_{ij}^{(t)}) \left( \mathbf{W}_{ij}^{(t+1)} + \mathbf{V}_{ij}^{(t)}/\rho_t \right)^2 \qquad (33) \\
&= \left\| \left( \mathbf{W}^{(t+1)} + \mathbf{V}^{(t)}/\rho_t \right) \odot (1 - \mathbf{S}^{(t)}) \right\|_F^2 \\
&\leq \left\| \mathbf{D}^{(t)} - \left( \mathbf{W}^{(t+1)} + \mathbf{V}^{(t)}/\rho_t \right) \right\|_F^2 .
\end{aligned}
$$

Here the first equality follows from the representation $\mathbf{D}^{(t+1)} = \left( \mathbf{W}^{(t+1)} + \mathbf{V}^{(t)}/\rho_t \right) \odot \mathbf{S}^{(t+1)}$, the first inequality follows directly from Assumption 1, and the final inequality leverages the fact that $\mathbf{D}_{ij}^{(t)} = 0$ if $\mathbf{S}_{ij}^{(t)} = 0$.

Having established inequality (32), all remaining components of the convergence proof from [Meng et al., 2024a, Theorem 1] apply directly to our setting. Therefore, the sequences $\{\mathbf{D}^{(t)}\}_{t=0}^{\infty}$ and $\{\mathbf{W}^{(t)}\}_{t=0}^{\infty}$ converge to a common limit $\bar{\mathbf{W}}$, completing the proof.

### A.2 Algorithmic implementation details

In this section, we present our tensor-based implementation of Algorithms 1 and 2, which enables parallel processing of millions of weight blocks simultaneously on GPUs, yielding significant computational speedup.

Our implementation is based on PyTorch. Given a weight matrix $\mathbf{W}$ and the desired transposable N:M sparsity parameters, we first reshape the matrix into a tensor of shape $(B, M, M)$, where $B$ represents the number of $M \times M$ blocks. All algorithmic operations are then applied simultaneously to this batched tensor representation.

**Implementation of Algorithm 1**  The Dykstra algorithm naturally lends itself to tensor-based operations as it primarily involves matrix-vector multiplications and element-wise operations. These operations are directly parallelizable across all blocks through PyTorch's built-in broadcasting mechanisms. A critical implementation detail is performing all operations in log-space to ensure numerical stability, particularly when using large entropy regularization parameters $\tau$:

```python
def log_softmax_normalize(x, dim, N):
    # Stable log-domain normalization
    lse = torch.logsumexp(x, dim=dim, keepdim=True)
    return x - (lse - torch.log(torch.tensor(N)))

# Batched Dykstra projections in log-space
log_S = tau * torch.abs(batch_W)
for _ in range(max_iter):
    # Projection 1: Row marginal (sum_i exp(log_S) = N)
    log_S = log_softmax_normalize(log_S, dim=1, N=N)

    # Projection 2: Column marginal (sum_j exp(log_S) = N)
    log_S = log_softmax_normalize(log_S, dim=2, N=N)

    # Projection 3: Capacity constraint (S <= 1)
    log_tmp = log_S + log_Q
    log_S = torch.minimum(log_tmp, torch.zeros_like(log_tmp))
    log_Q = log_tmp - log_S  # Dual variable update
```

**Implementation of Algorithm 2** Vectorizing the rounding procedure presents greater challenges due to its conditional logic and iterative selection process. We overcome these challenges through careful masking and parallel accumulation operations. The key insight is transforming conditional logic into parallel tensor operations that work across all blocks simultaneously. For the greedy selection phase, we pre-compute all sorted indices and then use Boolean masks and accumulation counters to track row and column capacities across all blocks in parallel. For the local search phase, we similarly transform the conditional swap operations into tensor-based score computation and masked updates, enabling simultaneous identification of deficit rows/columns and optimal swap positions across all blocks.

```python
# Vectorized greedy selection
sorted_idx = abs_blocks.flatten(1).argsort(dim=1, descending=True)
rows, cols = sorted_idx // M, sorted_idx % M
row_counts, col_counts = torch.zeros((B, M)),torch.zeros((B, M))
mask = torch.zeros((B, M, M), dtype=torch.bool, device=device)
batch_idx = torch.arange(B, device=device)

for k in range(M*M):
    r, c = rows[:, k], cols[:, k]
    can_select = (row_counts[batch_idx, r] < N) & \
                 (col_counts[batch_idx, c] < N)

    # Update mask and counters for all blocks simultaneously
    mask[batch_idx, r, c] |= can_select
    row_counts[batch_idx, r] += can_select
    col_counts[batch_idx, c] += can_select

# Vectorized local search
for _ in range(num_iter):
    # Find unsaturated rows and columns
    row_deficit = mask.sum(dim=2) < N  # Shape: (B, M)
    col_deficit = mask.sum(dim=1) < N  # Shape: (B, M)
    needs_fix = row_deficit.any(dim=1) | col_deficit.any(dim=1)

    # Select deficit rows/columns (according to some criteria)
    row_idx = select_deficit(row_deficit)
    col_idx = select_deficit(col_deficit)

    # Compute and apply optimal swaps across all blocks
    score = compute_swap_scores(W, mask, row_idx, col_idx)
    max_score, flat_idx = score.flatten(1).max(dim=1)
    swap_valid = (max_score > 0) & needs_fix

    # Apply swaps only to blocks needing improvement
    update_blocks = torch.where(swap_valid)[0]
    apply_swaps(mask, update_blocks, row_idx, col_idx, flat_idx)
```

# B  Experimental Details

## B.1  Experimental setup

**Computing environments** All experiments were conducted on a computing cluster. Unless otherwise specified, we utilized an Intel Xeon Gold 6248 machine with 20 CPU cores and a single NVIDIA A100 GPU, featuring 192GB of system RAM and 40GB of GPU memory. All language models and pruning methods were implemented using the PyTorch library Paszke et al. [2017].

**Choice of parameters** For Algorithm 1, we set the regularization parameter $\tau$ to $0.005 \max_{ij} |\mathbf{W}_{ij}|$ and limit the maximum iterations to $T = 300$. In Algorithm 2, we perform $L = 10$ local search steps to refine the solution.

**Implementation Details** We provide configuration and implementation specifications for both baseline methods and integration frameworks utilized in our comparative analysis of transposable N:M sparsity solvers. For baseline methods:

- **Network Flow:** We utilize the official implementation from Hubara et al. [2021a] (accessible via GitHub) and adapt it for transposable N:M mask generation. To optimize computational efficiency, we employ CPU multi-processing with 16 parallel threads.
- **2-Approximation:** We utilize the official implementation from Hubara et al. [2021a] (accessible via GitHub) and adapt it for transposable N:M mask generation. Similarly, we maximize throughput via CPU multi-processing with 16 parallel threads.
- **Bi-NM:** We adapt the method from [Zhang et al., 2023] with slight modifications. First, we apply magnitude-based row-wise N:M sparsity to weight matrix $\mathbf{W}$, generating mask $\mathbf{S}_1$ such that $\mathbf{W} \odot \mathbf{S}_1$ satisfies row-wise N:M sparsity. Subsequently, we impose column-wise N:M sparsity on $\mathbf{W} \odot \mathbf{S}_1$ to obtain mask $\mathbf{S}_2$. The composite mask $\mathbf{S}_1 \odot \mathbf{S}_2$ then satisfies the transposable N:M sparsity requirement.
- **cuPDLP:** We employ the official Julia implementation from [Lu and Yang, 2023] (accessible via GitHub). For this method, we reformulate problem (1) as a linear programming problem by relaxing the binary constraint $S_{ij} \in \{0, 1\}$ to $S_{ij} \in [0, 1]$. This relaxed formulation is then processed by cuPDLP. Notably, we apply cuPDLP directly to the entire weight matrix rather than partitioning it into multiple $M \times M$ submatrices (i.e., solving (3) block by block), as the latter approach would significantly inhibit GPU acceleration capabilities.

Below are the implementation specifications for the N:M pruning frameworks with which we evaluated and integrated our proposed solver *TSENOR*:

- **SparseGPT:** We adopt the official implementation from Frantar and Alistarh [2023] (accessible via GitHub) and apply the default hyperparameters for one-shot N:M pruning of LLMs.
- **Magnitude Pruning:** We implement magnitude-based pruning by directly applying our proposed solver *TSENOR* to the weight matrices at each network layer to generate transposable N:M masks.
- **Wanda:** We utilize the official implementation from Sun et al. [2023] (accessible via GitHub) with default hyperparameters for one-shot transposable N:M pruning of LLMs. The detailed integration procedure of our solver with Wanda is elaborated in Section 4.
- **ALPS:** We employ the official implementation from Meng et al. [2024a] (accessible via GitHub) with default hyperparameters for one-shot (transposable) N:M pruning of LLMs. The detailed integration procedure of our solver with ALPS is elaborated in Section 4.

**Finetuning and Retraining Details** We adopt the official implementation of Bi-NM Zhang et al. [2023](accessible via GitHub), and adapt it for retraining pruned LLMs. To ensure a fair comparison, we avoid sparse training from scratch. Instead, we first apply magnitude-based N:M pruning to pre-trained LLaMA3.2-1B, and then retrain the resulting model using Bi-NM.

For both finetuning and Bi-NM retraining, we use a learning rate of 2e-5 and a batch size of size 64 per step. The block size used is 1024 tokens per batch. The effective batch size is obtained by using a physical batch size of 2 on GPU with 32 gradient accumulation steps before each weight update. Training is conducted on the first shard of the C4 training dataset, which contains over 150 million tokens. We employ the Adam optimizer with PyTorch's default hyperparameters. A cosine learning rate scheduler is used, with a warmup ratio of 0.03 and no weight decay applied.

## B.2 Ablation studies and additional results

### B.2.1 Effectiveness of entropy regularization and rounding procedures

We evaluate the individual contributions of each component in our proposed solver *TSENOR* for generating high-quality transposable N:M masks. Our analysis compares three distinct rounding strategies:

- **Simple:** Sequential application of row-wise N:M sparse rounding followed by column-wise N:M sparse rounding to obtain the transposable N:M mask.
- **Greedy:** Implementation of only the greedy selection procedure for rounding (lines 1-8 in Algorithm 2).
- **Optround:** Our complete proposed rounding approach detailed in Algorithm 2, incorporating both greedy selection and local search optimization.

For each strategy, we evaluate performance when rounding the approximate mask generated by our entropy-regularized optimization (Algorithm 1). As a baseline comparison, we also apply each rounding procedure directly to the magnitude of matrix weights (i.e., $|\mathbf{W}|$). Figure 6 presents these experimental results.

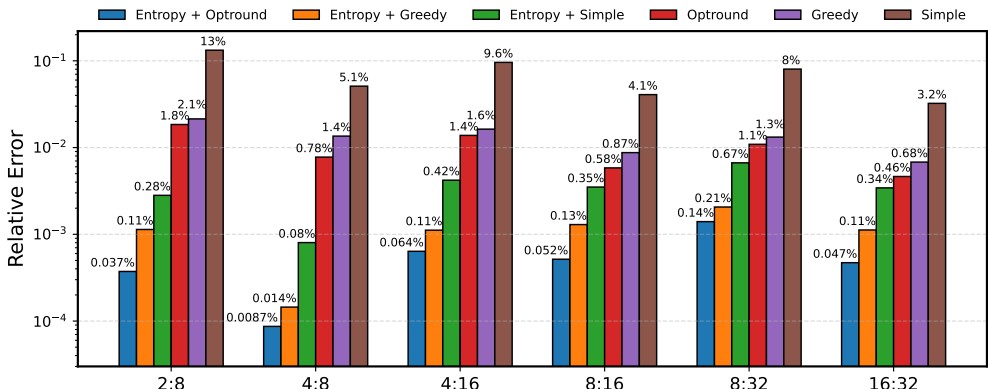

Figure 6: Solution quality comparison for transposable N:M mask generation. For various N:M sparsity patterns, we evaluate all methods on 100 M×M blocks sampled from LLaMA3 [Dubey et al., 2024] model weights. We report the average relative error, defined as $(f(\mathbf{S}^*) - f(\mathbf{S}))/f(\mathbf{S}^*)$, where $\mathbf{S}^*$ is the optimal support and $f$ is the objective defined in (1). "Entropy+" indicates rounding applied to masks generated by Algorithm 1.

Our findings demonstrate that each component of our proposed rounding procedure contributes significantly to mask quality improvement. The greedy selection step reduces error by 50-90%, while the subsequent local search optimization further reduces error by up to an additional 50%. Moreover, applying our rounding techniques to the approximate mask generated through entropy regularization yields solutions with less than 5% error compared to direct application to the magnitude of weight matrix $|\mathbf{W}|$. These results conclusively validate the importance of each component in our proposed methodology.

### B.2.2   Acceleration from exploiting GPU parallelism

We assess the computational gains from GPU acceleration and algorithm vectorization in our transposable N:M sparsity solver. Table 3 compares runtime performance of our Dykstra method (Algorithm 1) and rounding procedure (Algorithm 2) across CPU (direct and vectorized) and GPU implementations.

| Matrix Size | Dykstra method (Algorithm 1) | | | | Rounding (Algorithm 2) | | | | |
|---|---|---|---|---|---|---|---|---|---|
| | CPU | V100 | A100 | H100 | CPU | CPU(V) | V100 | A100 | H100 |
| $512 \times 512$ | 1.29 (±0.02) | 0.04 (±0.00) | 0.04 (±0.00) | 0.03 (±0.00) | 1.01 (±0.01) | 0.21 (±0.00) | 0.07 (±0.00) | 0.10 (±0.00) | 0.05 (±0.00) |
| $2048 \times 2048$ | 21.1 (±0.24) | 0.16 (±0.00) | 0.10 (±0.00) | 0.07 (±0.00) | 16.5 (±0.17) | 2.87 (±0.02) | 0.11 (±0.00) | 0.10 (±0.00) | 0.05 (±0.00) |
| $8192 \times 8192$ | 344 (±2.32) | 2.99 (±0.00) | 1.60 (±0.00) | 0.97 (±0.00) | 270 (±1.30) | 38.2 (±0.15) | 0.27 (±0.00) | 0.14 (±0.00) | 0.09 (±0.00) |

Table 3: Runtime (seconds) for transposable 8:16 sparsity tested on four different device: a single Intel Xeon Gold 6248 CPU, NVIDIA V100-PCIe-32GB, A100-PCIe-40GB, and H100-PCIe-80GB. CPU(V) denotes vectorized implementation. Results are averaged over 10 trials with standard deviations in parentheses.

Our hardware-aware design yields remarkable speedups: GPU acceleration provides up to 300× faster execution for the Dykstra method compared to CPU. For the rounding procedure, our vectorized CPU implementation achieves an 8× speedup, while GPU acceleration delivers up to 3000× acceleration over the baseline CPU implementation. These results demonstrate how our tensor operations based algorithms effectively leverage GPU architectures, particularly for large matrices (8192 × 8192), enabling efficient processing of state-of-the-art models with minimal computational overhead.

### B.2.3 Layer-wise reconstruction error comparison

We investigate the impact of transposable N:M sparsity on model performance by analyzing layer-wise reconstruction error in LLaMA3.2-8B after applying the ALPS one-shot pruning method. We define reconstruction error as $\|\mathbf{X}\widehat{\mathbf{W}} - \mathbf{X}\mathbf{W}\|_F^2/\|\mathbf{X}\widehat{\mathbf{W}}\|_F^2$. Table 4 presents results for various N:M configurations.

Our analysis reveals that while transposable N:M sparsity increases reconstruction error compared to standard N:M sparsity, this differential diminishes with larger M values—decreasing from approximately 30% additional error at small M values to only 7% at larger M values. Notably, at M=32, the additional error introduced by transposable constraints over standard N:M sparsity is smaller than the error increment from unstructured to standard N:M sparsity. Furthermore, transposable N:M sparsity with M=32 produces substantially lower reconstruction error than standard N:M sparsity with M=4. These findings validate that N:M patterns with larger M values offer superior performance and underscore the importance of our proposed solver.

| Sparsity | 50% (uns: 0.021) | | | | 62.5% (uns: 0.045) | | |
| | 2:4 | 4:8 | 8:16 | 16:32 | 3:8 | 6:16 | 12:32 |
|---|---|---|---|---|---|---|---|
| N:M | 0.041 | 0.031 | 0.026 | 0.024 | 0.063 | 0.055 | 0.050 |
| Tran N:M | 0.058 | 0.041 | 0.032 | 0.027 | 0.078 | 0.063 | 0.056 |
| Sparsity | 75% (uns: 0.093) | | | | 87.5% (uns: 0.199) | | |
| | 1:4 | 2:8 | 4:16 | 8:32 | 1:8 | 2:16 | 4:32 |
| N:M | 0.143 | 0.121 | 0.108 | 0.102 | 0.247 | 0.227 | 0.215 |
| Tran N:M | 0.184 | 0.146 | 0.122 | 0.109 | 0.291 | 0.253 | 0.229 |

Table 4: Layer-wise reconstruction error for the "self_attn.k_proj" layer in the first block of LLaMA3-8B, comparing standard N:M and transposable N:M sparsity across different N:M patterns. The reconstruction error values for unstructured pruning at each sparsity level are shown in parentheses.

### B.2.4 Comparing standard and transposable N:M sparsity after fine-tuning

We have compared one-shot pruning performance between transposable and standard N:M sparsity in Figure 4. We now compare fine-tuning performance between these settings across different N:M patterns on LLaMA3.2-1B in the following table. Similar to one-shot pruning, the performance gap between transposable N:M and standard N:M diminishes dramatically as M increases from 4 to 32 (by approximately 80%).

| Sparsity | 2:4 | 4:8 | 8:16 | 16:32 |
|---|---|---|---|---|
| N:M | 19.4 | 18.1 | 17.4 | 17.1 |
| Tran N:M | 22.1 | 19.7 | 18.4 | 17.7 |

Table 5: Perplexity comparison of LLaMA3.2-1B pruned using ALPS with *TSENOR* followed by fine-tuning, evaluated on both standard and transposable N:M sparsity across various N:M patterns.

### B.2.5 Comprehensive model performance across N:M sparsity patterns

We evaluate the performance of one-shot pruning methods by comparing SparseGPT, ALPS, and *TSENOR* integrated with various techniques (MP, Wanda, and ALPS) on LLaMA models, as presented in Tables 6-8. Our experiments examine both standard N:M and transposable N:M sparsity patterns across multiple configurations (1:4, 2:8, 4:16, 8:32, 2:4, 4:8, 8:16, and 16:32). Performance evaluation contains perplexity on WikiText2, PTB, and C4 datasets, as well as reasoning abilities assessed through accuracy on PIQA, ARC-Easy (ARC-E), ARC-Challenge (ARC-C), Hellaswag (HS), Winogrande (WG), RTE, OpenbookQA (OQA), and BoolQ.

| N:M Sparsity | Algorithm | Transpose | Perplexity (↓) | | | Zero-shot (↑) | | | | | | | | |
|---|---|---|---|---|---|---|---|---|---|---|---|---|---|---|
| | | | C4 | WT2 | PTB | PIQA | HS | ARC-E | ARC-C | WG | RTE | OQA | BoolQ | Avg |
| 1:4 | SparseGPT | ✗ | 738.04 | 1103.67 | 977.13 | 50.82 | 26.23 | 28.20 | 23.38 | 51.22 | 50.54 | 26.80 | 38.87 | 37.01 |
| | ALPS | ✗ | 203.17 | 236.28 | 284.55 | 54.30 | 27.77 | 29.63 | 21.59 | 50.83 | 54.15 | 26.40 | 42.81 | 38.44 |
| | TSENOR+Wanda | ✓ | 264880.44 | 379528.66 | 332039.06 | 50.05 | 26.67 | 25.29 | 26.19 | 49.72 | 47.29 | 30.60 | 38.59 | 36.80 |
| | TSENOR+SparseGPT | ✓ | 3812.92 | 8804.32 | 8825.35 | 51.03 | 25.53 | 25.80 | 26.11 | 50.12 | 52.35 | 28.40 | 38.26 | 37.20 |
| | TSENOR+ALPS | ✓ | 788.83 | 1032.93 | 1570.19 | 52.29 | 25.83 | 27.36 | 25.09 | 48.22 | 51.99 | 26.40 | 38.81 | 37.00 |
| 2:8 | SparseGPT | ✗ | 377.12 | 474.92 | 457.05 | 52.94 | 27.84 | 30.60 | 21.84 | 49.25 | 52.35 | 25.20 | 43.67 | 37.96 |
| | ALPS | ✗ | 173.36 | 161.46 | 195.52 | 56.37 | 28.67 | 32.28 | 21.59 | 51.78 | 52.71 | 25.80 | 46.82 | 39.50 |
| | TSENOR+Wanda | ✓ | 1144164.00 | 1047911.25 | 593504.12 | 51.58 | 25.75 | 25.80 | 26.14 | 49.49 | 48.74 | 28.20 | 52.75 | 38.62 |
| | TSENOR+SparseGPT | ✓ | 1931.98 | 3395.00 | 4627.83 | 52.50 | 26.07 | 27.74 | 22.78 | 48.93 | 50.90 | 28.00 | 40.95 | 37.23 |
| | TSENOR+ALPS | ✓ | 363.74 | 472.95 | 563.83 | 52.99 | 27.24 | 29.67 | 21.50 | 48.78 | 53.79 | 26.40 | 41.01 | 37.67 |
| 4:16 | SparseGPT | ✗ | 253.21 | 246.20 | 230.20 | 54.19 | 27.97 | 30.51 | 20.73 | 48.93 | 52.71 | 26.40 | 49.88 | 38.92 |
| | ALPS | ✗ | 149.47 | 132.16 | 184.76 | 55.77 | 28.65 | 31.94 | 20.90 | 52.49 | 51.99 | 25.00 | 51.01 | 39.72 |
| | TSENOR+Wanda | ✓ | 128702.87 | 118177.39 | 77005.69 | 51.25 | 26.71 | 24.75 | 25.60 | 49.96 | 48.38 | 28.40 | 45.23 | 37.53 |
| | TSENOR+SparseGPT | ✓ | 906.56 | 1434.63 | 1882.64 | 52.88 | 26.47 | 28.16 | 22.95 | 49.49 | 51.99 | 24.60 | 39.72 | 37.03 |
| | TSENOR+ALPS | ✓ | 218.61 | 256.94 | 298.77 | 53.48 | 27.83 | 29.63 | 21.33 | 50.91 | 52.35 | 26.80 | 42.57 | 38.11 |
| 8:32 | SparseGPT | ✗ | 228.11 | 227.82 | 292.42 | 54.90 | 27.94 | 30.68 | 22.18 | 49.64 | 52.71 | 25.80 | 61.04 | 40.61 |
| | ALPS | ✗ | 128.57 | 116.78 | 140.46 | 55.66 | 28.86 | 32.07 | 21.33 | 51.85 | 52.71 | 25.80 | 60.67 | 41.12 |
| | TSENOR+Wanda | ✓ | 142482.83 | 109316.03 | 76792.27 | 51.74 | 26.88 | 25.42 | 26.71 | 51.46 | 52.71 | 29.40 | 46.73 | 38.88 |
| | TSENOR+SparseGPT | ✓ | 605.25 | 889.53 | 1389.67 | 54.79 | 26.89 | 30.60 | 23.04 | 51.07 | 51.62 | 27.20 | 38.17 | 37.92 |
| | TSENOR+ALPS | ✓ | 169.09 | 176.82 | 217.66 | 53.10 | 27.93 | 30.98 | 22.10 | 50.83 | 53.07 | 28.00 | 41.25 | 38.41 |
| 2:4 | SparseGPT | ✗ | 44.96 | 32.69 | 50.79 | 62.57 | 39.18 | 41.92 | 23.63 | 54.93 | 54.51 | 29.20 | 62.17 | 46.01 |
| | ALPS | ✗ | 35.14 | 26.50 | 41.13 | 63.60 | 41.92 | 44.11 | 25.68 | 55.41 | 52.71 | 30.60 | 62.32 | 47.04 |
| | TSENOR+Wanda | ✓ | 12718.35 | 32041.32 | 30749.90 | 52.67 | 27.19 | 27.78 | 23.63 | 50.36 | 52.71 | 24.20 | 41.99 | 37.56 |
| | TSENOR+SparseGPT | ✓ | 93.09 | 87.79 | 106.94 | 56.58 | 31.33 | 34.81 | 21.50 | 52.01 | 53.79 | 26.20 | 60.70 | 42.12 |
| | TSENOR+ALPS | ✓ | 56.11 | 51.11 | 75.79 | 59.09 | 34.89 | 36.91 | 23.29 | 50.91 | 53.07 | 27.00 | 62.14 | 43.41 |
| 4:8 | SparseGPT | ✗ | 33.33 | 24.77 | 38.71 | 65.56 | 43.37 | 44.87 | 25.85 | 54.30 | 53.79 | 31.40 | 62.17 | 47.66 |
| | ALPS | ✗ | 27.56 | 20.56 | 32.89 | 68.01 | 46.91 | 48.86 | 28.50 | 56.75 | 52.71 | 32.80 | 62.17 | 49.59 |
| | TSENOR+Wanda | ✓ | 1226.75 | 1293.31 | 1546.12 | 54.13 | 28.50 | 31.23 | 21.76 | 49.33 | 52.71 | 23.40 | 40.73 | 37.72 |
| | TSENOR+SparseGPT | ✓ | 52.08 | 42.09 | 60.47 | 60.99 | 37.59 | 39.06 | 25.17 | 53.04 | 53.79 | 28.00 | 61.01 | 44.83 |
| | TSENOR+ALPS | ✓ | 37.94 | 29.76 | 47.25 | 64.36 | 41.08 | 40.82 | 25.77 | 55.41 | 52.35 | 28.60 | 62.14 | 46.32 |
| 8:16 | SparseGPT | ✗ | 28.02 | 20.27 | 33.15 | 67.03 | 47.26 | 48.15 | 27.82 | 55.41 | 52.71 | 31.80 | 62.32 | 49.06 |
| | ALPS | ✗ | 24.22 | 17.77 | 28.78 | 67.57 | 49.87 | 49.96 | 28.16 | 57.06 | 52.71 | 31.80 | 63.52 | 50.08 |
| | TSENOR+Wanda | ✓ | 610.70 | 483.12 | 446.68 | 56.64 | 30.01 | 33.33 | 21.76 | 50.75 | 54.87 | 22.20 | 39.94 | 38.69 |
| | TSENOR+SparseGPT | ✓ | 39.72 | 31.93 | 51.89 | 63.44 | 40.57 | 43.86 | 24.32 | 54.30 | 54.87 | 28.20 | 61.87 | 46.43 |
| | TSENOR+ALPS | ✓ | 30.38 | 22.82 | 36.23 | 66.10 | 44.75 | 47.01 | 27.47 | 56.51 | 53.07 | 30.20 | 62.14 | 48.41 |
| 16:32 | SparseGPT | ✗ | 26.91 | 19.13 | 32.14 | 66.97 | 48.45 | 48.19 | 27.47 | 55.88 | 53.43 | 33.00 | 62.26 | 49.46 |
| | ALPS | ✗ | 22.97 | 16.75 | 27.61 | 68.88 | 51.30 | 50.34 | 30.03 | 56.43 | 52.71 | 32.80 | 62.94 | 50.68 |
| | TSENOR+Wanda | ✓ | 144.48 | 115.23 | 173.61 | 61.21 | 36.12 | 38.72 | 23.89 | 51.54 | 51.26 | 25.20 | 47.19 | 41.89 |
| | TSENOR+SparseGPT | ✓ | 31.96 | 24.16 | 40.13 | 64.53 | 44.38 | 46.63 | 26.11 | 56.04 | 52.71 | 29.60 | 62.11 | 47.76 |
| | TSENOR+ALPS | ✓ | 25.92 | 19.12 | 31.08 | 66.21 | 47.88 | 50.55 | 29.95 | 55.80 | 54.51 | 30.20 | 62.39 | 49.69 |

Table 6: Performance analysis for (transposable) N:M pruning on LLaMA3.2-1B model. Lower values are preferred for perplexity, and higher values are preferred for zero-shot tasks.

| N:M Sparsity | Algorithm | Transpose | Perplexity (↓) | | | Zero-shot (↑) | | | | | | | | |
|---|---|---|---|---|---|---|---|---|---|---|---|---|---|---|
| | | | C4 | WT2 | PTB | PIQA | HS | ARC-E | ARC-C | WG | RTE | OQA | BoolQ | Avg |
| 1:4 | SparseGPT | ✗ | 287.89 | 303.27 | 389.16 | 52.77 | 28.17 | 28.32 | 23.21 | 48.78 | 52.71 | 27.20 | 37.83 | 37.37 |
| | ALPS | ✗ | 142.86 | 227.03 | 246.41 | 54.08 | 27.79 | 28.62 | 21.59 | 50.28 | 51.99 | 26.60 | 57.22 | 39.77 |
| | TSENOR+Wanda | ✓ | 217413.53 | 216446.80 | 234474.44 | 51.14 | 26.17 | 25.55 | 27.47 | 52.80 | 53.07 | 27.40 | 46.94 | 38.82 |
| | TSENOR+SparseGPT | ✓ | 838.06 | 2068.74 | 3004.19 | 52.01 | 26.35 | 27.10 | 25.68 | 49.25 | 51.62 | 28.00 | 37.83 | 37.23 |
| | TSENOR+ALPS | ✓ | 295.36 | 526.59 | 599.66 | 53.10 | 27.15 | 28.07 | 22.78 | 50.20 | 52.35 | 27.60 | 37.89 | 37.39 |
| 2:8 | SparseGPT | ✗ | 196.82 | 238.63 | 295.95 | 53.21 | 28.28 | 29.38 | 21.84 | 49.01 | 51.26 | 26.80 | 51.65 | 38.93 |
| | ALPS | ✗ | 103.22 | 119.57 | 163.66 | 54.73 | 29.15 | 30.18 | 20.82 | 48.46 | 52.71 | 26.00 | 61.87 | 40.49 |
| | TSENOR+Wanda | ✓ | 163461.16 | 170679.30 | 192995.58 | 51.14 | 26.14 | 25.00 | 25.00 | 50.04 | 53.79 | 27.60 | 45.23 | 38.37 |
| | TSENOR+SparseGPT | ✓ | 452.98 | 789.27 | 991.03 | 52.07 | 26.85 | 28.16 | 23.55 | 49.72 | 51.99 | 30.40 | 37.83 | 37.57 |
| | TSENOR+ALPS | ✓ | 189.88 | 238.85 | 288.52 | 53.10 | 27.53 | 28.83 | 21.76 | 49.49 | 52.71 | 27.20 | 41.74 | 37.80 |
| 4:16 | SparseGPT | ✗ | 151.83 | 151.59 | 206.49 | 53.32 | 28.49 | 29.76 | 21.67 | 48.70 | 56.68 | 28.60 | 61.19 | 41.05 |
| | ALPS | ✗ | 84.13 | 95.39 | 121.15 | 55.82 | 30.36 | 31.94 | 20.05 | 50.67 | 52.71 | 26.80 | 62.14 | 41.31 |
| | TSENOR+Wanda | ✓ | 60623.27 | 94040.87 | 120817.32 | 50.60 | 25.84 | 24.75 | 27.05 | 49.64 | 50.54 | 27.80 | 54.25 | 38.81 |
| | TSENOR+SparseGPT | ✓ | 327.64 | 395.35 | 571.11 | 52.39 | 27.56 | 29.21 | 23.81 | 47.99 | 52.71 | 25.40 | 46.67 | 38.22 |
| | TSENOR+ALPS | ✓ | 137.25 | 186.89 | 226.89 | 53.97 | 28.11 | 30.39 | 21.59 | 50.99 | 52.71 | 27.20 | 55.41 | 40.05 |
| 8:32 | SparseGPT | ✗ | 139.67 | 120.22 | 183.01 | 53.86 | 28.80 | 29.50 | 21.67 | 48.46 | 54.15 | 27.20 | 62.23 | 40.74 |
| | ALPS | ✗ | 80.12 | 82.28 | 110.73 | 56.64 | 31.27 | 32.66 | 19.80 | 51.38 | 52.35 | 27.00 | 61.99 | 41.63 |
| | TSENOR+Wanda | ✓ | 73379.13 | 100992.48 | 289678.69 | 50.98 | 26.44 | 25.84 | 27.99 | 50.12 | 52.71 | 26.60 | 55.26 | 39.32 |
| | TSENOR+SparseGPT | ✓ | 239.08 | 302.73 | 378.69 | 52.56 | 27.33 | 28.62 | 23.04 | 49.80 | 52.71 | 26.40 | 41.35 | 37.72 |
| | TSENOR+ALPS | ✓ | 111.36 | 163.18 | 178.60 | 54.90 | 28.92 | 30.43 | 20.39 | 50.99 | 53.07 | 26.60 | 61.22 | 40.81 |
| 2:4 | SparseGPT | ✗ | 28.30 | 21.61 | 34.47 | 69.10 | 50.45 | 51.64 | 28.92 | 60.54 | 52.35 | 30.80 | 68.62 | 51.55 |
| | ALPS | ✗ | 23.92 | 18.76 | 27.28 | 70.62 | 54.43 | 56.19 | 29.35 | 60.54 | 52.35 | 33.00 | 68.50 | 53.12 |
| | TSENOR+Wanda | ✓ | 5433.63 | 5336.60 | 4977.34 | 51.47 | 26.89 | 27.74 | 23.21 | 51.07 | 52.71 | 26.60 | 37.80 | 37.18 |
| | TSENOR+SparseGPT | ✓ | 56.12 | 45.49 | 87.28 | 61.81 | 36.78 | 40.99 | 24.23 | 51.78 | 52.71 | 27.40 | 62.20 | 44.74 |
| | TSENOR+ALPS | ✓ | 41.19 | 35.02 | 54.76 | 63.93 | 42.47 | 44.49 | 26.96 | 54.30 | 52.71 | 28.40 | 62.87 | 47.02 |
| 4:8 | SparseGPT | ✗ | 21.71 | 16.06 | 24.90 | 70.46 | 57.57 | 57.41 | 32.51 | 63.61 | 53.07 | 33.80 | 70.46 | 54.86 |
| | ALPS | ✗ | 19.43 | 14.54 | 22.29 | 71.38 | 59.37 | 60.40 | 36.01 | 65.35 | 59.21 | 35.40 | 69.30 | 57.05 |
| | TSENOR+Wanda | ✓ | 1142.41 | 1184.56 | 974.31 | 54.03 | 27.75 | 30.64 | 22.95 | 50.59 | 52.71 | 26.00 | 41.50 | 38.27 |
| | TSENOR+SparseGPT | ✓ | 31.15 | 24.33 | 36.81 | 67.79 | 49.37 | 50.84 | 28.24 | 58.33 | 57.04 | 31.00 | 65.38 | 51.00 |
| | TSENOR+ALPS | ✓ | 25.65 | 20.11 | 29.75 | 69.37 | 52.49 | 55.09 | 29.27 | 59.19 | 57.76 | 33.00 | 67.16 | 52.92 |
| 8:16 | SparseGPT | ✗ | 19.21 | 13.99 | 22.00 | 73.07 | 60.56 | 61.83 | 34.30 | 63.06 | 58.12 | 34.60 | 71.35 | 57.11 |
| | ALPS | ✗ | 17.60 | 12.92 | 20.02 | 73.12 | 62.38 | 64.14 | 37.88 | 66.30 | 55.60 | 36.40 | 72.84 | 58.58 |
| | TSENOR+Wanda | ✓ | 966.21 | 686.54 | 729.43 | 55.98 | 29.28 | 34.72 | 22.44 | 49.33 | 52.71 | 26.00 | 48.10 | 39.82 |
| | TSENOR+SparseGPT | ✓ | 23.92 | 17.75 | 27.56 | 70.24 | 55.56 | 56.23 | 31.48 | 59.75 | 55.60 | 32.80 | 65.69 | 53.42 |
| | TSENOR+ALPS | ✓ | 20.28 | 15.11 | 22.97 | 72.20 | 58.54 | 60.23 | 35.32 | 61.64 | 51.99 | 35.40 | 70.03 | 55.67 |
| 16:32 | SparseGPT | ✗ | 18.11 | 13.02 | 20.37 | 73.12 | 62.27 | 62.67 | 34.81 | 65.75 | 60.65 | 35.20 | 71.53 | 58.25 |
| | ALPS | ✗ | 16.74 | 12.06 | 18.78 | 73.56 | 64.10 | 64.48 | 36.52 | 66.54 | 57.40 | 39.00 | 72.81 | 59.30 |
| | TSENOR+Wanda | ✓ | 436.13 | 262.78 | 363.91 | 57.29 | 30.54 | 37.79 | 21.25 | 50.28 | 52.71 | 25.60 | 49.82 | 40.66 |
| | TSENOR+SparseGPT | ✓ | 20.29 | 14.80 | 23.81 | 72.36 | 58.69 | 59.64 | 33.02 | 62.06 | 53.79 | 34.20 | 65.99 | 55.10 |
| | TSENOR+ALPS | ✓ | 18.03 | 13.08 | 20.43 | 72.96 | 61.43 | 63.22 | 38.05 | 63.46 | 57.40 | 35.60 | 72.32 | 58.06 |

Table 7: Performance analysis for (transposable) N:M pruning on LLaMA3.2-3B model. Lower values are preferred for perplexity, and higher values are preferred for zero-shot tasks.

| N:M Sparsity | Algorithm | Transpose | Perplexity (↓) | | | Zero-shot (↑) | | | | | | | | |
|---|---|---|---|---|---|---|---|---|---|---|---|---|---|---|
| | | | C4 | WT2 | PTB | PIQA | HS | ARC-E | ARC-C | WG | RTE | OQA | BoolQ | Avg |
| 1:4 | SparseGPT | ✗ | 188.81 | 221.53 | 317.36 | 52.45 | 28.07 | 29.17 | 21.33 | 47.91 | 54.15 | 26.80 | 37.98 | 37.23 |
| | ALPS | ✗ | 139.22 | 143.71 | 162.76 | 54.57 | 28.88 | 31.23 | 20.22 | 49.01 | 52.35 | 26.40 | 45.93 | 38.57 |
| | TSENOR+Wanda | ✓ | 1828046.38 | 2090725.00 | 2004626.25 | 50.16 | 26.21 | 25.21 | 25.17 | 49.49 | 53.43 | 27.80 | 45.60 | 37.88 |
| | TSENOR+SparseGPT | ✓ | 563.23 | 1116.69 | 1077.45 | 50.82 | 26.26 | 28.66 | 23.89 | 49.49 | 52.35 | 27.80 | 37.83 | 37.14 |
| | TSENOR+ALPS | ✓ | 213.25 | 339.98 | 354.03 | 52.77 | 27.27 | 29.42 | 20.82 | 49.17 | 52.35 | 27.20 | 37.83 | 37.10 |
| 2:8 | SparseGPT | ✗ | 130.02 | 158.07 | 182.28 | 53.54 | 28.65 | 29.92 | 23.04 | 49.80 | 53.07 | 27.60 | 40.52 | 38.27 |
| | ALPS | ✗ | 79.71 | 97.96 | 119.74 | 55.98 | 31.45 | 31.19 | 21.76 | 51.93 | 52.71 | 26.40 | 61.31 | 41.59 |
| | TSENOR+Wanda | ✓ | 337372.47 | 406156.78 | 230973.56 | 50.38 | 26.48 | 26.14 | 26.71 | 48.78 | 51.99 | 28.20 | 53.09 | 38.97 |
| | TSENOR+SparseGPT | ✓ | 272.81 | 406.95 | 414.92 | 52.34 | 27.18 | 28.62 | 21.93 | 48.70 | 52.71 | 26.40 | 37.83 | 36.96 |
| | TSENOR+ALPS | ✓ | 146.05 | 173.42 | 202.56 | 53.54 | 28.27 | 29.55 | 20.65 | 49.72 | 52.71 | 26.40 | 38.62 | 37.43 |
| 4:16 | SparseGPT | ✗ | 98.23 | 107.82 | 132.00 | 55.11 | 29.79 | 31.82 | 20.05 | 51.22 | 52.71 | 26.20 | 60.55 | 40.93 |
| | ALPS | ✗ | 65.58 | 64.25 | 92.76 | 57.83 | 33.26 | 33.25 | 20.99 | 54.14 | 53.79 | 27.20 | 62.14 | 42.83 |
| | TSENOR+Wanda | ✓ | 98070.10 | 103645.22 | 106283.87 | 51.52 | 26.25 | 25.67 | 26.37 | 50.36 | 51.99 | 30.40 | 37.89 | 37.56 |
| | TSENOR+SparseGPT | ✓ | 208.27 | 293.87 | 348.83 | 53.21 | 27.65 | 28.49 | 22.44 | 50.51 | 52.35 | 26.20 | 37.83 | 37.34 |
| | TSENOR+ALPS | ✓ | 111.71 | 140.05 | 166.55 | 54.13 | 29.21 | 30.18 | 21.16 | 50.67 | 52.71 | 26.60 | 43.58 | 38.53 |
| 8:32 | SparseGPT | ✗ | 90.09 | 89.71 | 115.57 | 56.09 | 30.60 | 31.44 | 20.90 | 51.62 | 52.71 | 28.60 | 60.15 | 41.51 |
| | ALPS | ✗ | 58.47 | 53.14 | 74.37 | 58.92 | 34.78 | 35.31 | 21.76 | 52.25 | 52.71 | 28.80 | 62.75 | 43.41 |
| | TSENOR+Wanda | ✓ | 79085.44 | 53720.92 | 37028.44 | 50.92 | 25.92 | 26.05 | 25.94 | 50.67 | 51.62 | 29.80 | 38.35 | 37.41 |
| | TSENOR+SparseGPT | ✓ | 170.92 | 214.02 | 317.44 | 53.05 | 27.32 | 28.75 | 22.70 | 49.33 | 52.71 | 28.80 | 39.02 | 37.71 |
| | TSENOR+ALPS | ✓ | 99.95 | 152.66 | 155.01 | 55.11 | 29.59 | 30.81 | 21.42 | 50.28 | 52.71 | 26.80 | 50.28 | 39.62 |
| 2:4 | SparseGPT | ✗ | 22.54 | 16.19 | 25.46 | 72.20 | 57.89 | 59.18 | 34.47 | 64.72 | 54.15 | 34.40 | 73.33 | 56.29 |
| | ALPS | ✗ | 19.67 | 14.61 | 22.18 | 73.83 | 61.67 | 61.11 | 35.92 | 67.17 | 56.32 | 34.80 | 73.52 | 58.04 |
| | TSENOR+Wanda | ✓ | 4010.53 | 4784.25 | 4851.96 | 52.72 | 27.51 | 29.59 | 23.21 | 50.28 | 52.71 | 26.20 | 38.44 | 37.58 |
| | TSENOR+SparseGPT | ✓ | 37.51 | 28.31 | 46.15 | 63.87 | 43.72 | 39.56 | 23.46 | 55.80 | 53.07 | 28.60 | 66.24 | 46.79 |
| | TSENOR+ALPS | ✓ | 32.65 | 24.74 | 39.00 | 67.85 | 48.20 | 48.32 | 28.75 | 59.98 | 52.71 | 29.20 | 66.18 | 49.91 |
| 4:8 | SparseGPT | ✗ | 17.50 | 12.31 | 18.55 | 74.37 | 65.40 | 62.42 | 38.14 | 69.30 | 55.23 | 37.20 | 75.08 | 59.64 |
| | ALPS | ✗ | 16.06 | 11.31 | 16.63 | 75.90 | 67.38 | 65.99 | 40.10 | 69.38 | 61.37 | 38.00 | 78.47 | 62.07 |
| | TSENOR+Wanda | ✓ | 1024.43 | 952.47 | 1390.95 | 54.95 | 29.70 | 34.09 | 22.35 | 50.99 | 53.07 | 27.20 | 39.33 | 38.96 |
| | TSENOR+SparseGPT | ✓ | 24.13 | 17.33 | 27.61 | 70.40 | 56.30 | 55.35 | 31.57 | 63.54 | 54.51 | 33.00 | 74.13 | 54.85 |
| | TSENOR+ALPS | ✓ | 21.79 | 15.77 | 23.34 | 73.01 | 59.98 | 61.45 | 34.39 | 66.38 | 57.76 | 34.40 | 69.94 | 57.16 |
| 8:16 | SparseGPT | ✗ | 15.48 | 10.66 | 16.04 | 75.63 | 68.41 | 65.74 | 40.70 | 69.69 | 59.93 | 39.00 | 77.55 | 62.08 |
| | ALPS | ✗ | 14.70 | 10.08 | 15.14 | 76.55 | 70.35 | 68.06 | 41.72 | 70.72 | 58.48 | 38.60 | 77.92 | 62.80 |
| | TSENOR+Wanda | ✓ | 383.74 | 291.99 | 413.10 | 57.78 | 32.44 | 37.42 | 22.35 | 52.41 | 52.35 | 27.40 | 44.04 | 40.77 |
| | TSENOR+SparseGPT | ✓ | 18.55 | 13.10 | 19.79 | 73.56 | 62.91 | 61.15 | 37.03 | 65.51 | 57.40 | 37.60 | 74.13 | 58.66 |
| | TSENOR+ALPS | ✓ | 17.03 | 11.96 | 17.98 | 75.90 | 66.24 | 66.62 | 38.31 | 67.96 | 56.68 | 36.60 | 77.37 | 60.71 |
| 16:32 | SparseGPT | ✗ | 14.72 | 9.92 | 15.41 | 76.71 | 69.66 | 68.69 | 42.15 | 71.27 | 59.93 | 39.60 | 76.82 | 63.10 |
| | ALPS | ✗ | 14.02 | 9.47 | 14.60 | 77.75 | 71.59 | 68.98 | 43.52 | 70.88 | 62.45 | 40.20 | 78.47 | 64.23 |
| | TSENOR+Wanda | ✓ | 338.95 | 219.25 | 379.07 | 61.04 | 34.11 | 44.82 | 25.00 | 52.96 | 52.71 | 28.80 | 55.17 | 44.33 |
| | TSENOR+SparseGPT | ✓ | 16.76 | 11.40 | 17.56 | 73.18 | 65.80 | 64.06 | 38.57 | 67.32 | 60.65 | 36.80 | 74.34 | 60.09 |
| | TSENOR+ALPS | ✓ | 15.11 | 10.28 | 15.74 | 76.50 | 68.64 | 68.81 | 41.13 | 67.80 | 61.37 | 39.00 | 79.11 | 62.80 |

Table 8: Performance analysis for (transposable) N:M pruning on LLaMA3.2-8B model. Lower values are preferred for perplexity, and higher values are preferred for zero-shot tasks.

### B.2.6 Additional results on fine-tuning transposable N:M sparse models

We show the results of fine-tuning transposable N:M sparse models pruned with our approach (TSENOR+ALPS) and compare against Bi-NM [Zhang et al., 2023] across multiple sparsity configurations (1:4, 2:8, 4:16, 8:32, 2:4, 4:8, 8:16, and 16:32). Performance evaluation includes perplexity measurements on WikiText2, PTB, and C4 datasets, as well as reasoning abilities assessed through accuracy scores on PIQA, ARC-Easy (ARC-E), ARC-Challenge (ARC-C), Hellaswag (HS), Winogrande (WG), RTE, OpenbookQA (OQA), and BoolQ.

| Method | Config | C4 ↓ | WT2 ↓ | PTB ↓ | PIQA ↑ | HS ↑ | ARC-E ↑ | ARC-C ↑ | WG ↑ | RTE ↑ | OQA ↑ | BoolQ ↑ | Avg ↑ |
|---|---|---|---|---|---|---|---|---|---|---|---|---|---|
| Bi-NM | 1:4 | 72.74 | 104.57 | 120.19 | 57.18 | 27.73 | 33.63 | 21.76 | 51.38 | 53.43 | 25.40 | 61.90 | 41.55 |
| TSENOR+ALPS | 1:4 | 48.84 | 57.66 | 75.04 | 58.16 | 28.28 | 34.51 | 22.61 | 52.41 | 53.43 | 24.40 | 62.08 | 41.99 |
| Bi-NM | 2:8 | 53.59 | 69.83 | 92.71 | 59.36 | 29.06 | 34.68 | 22.53 | 50.12 | 52.71 | 25.40 | 60.64 | 41.81 |
| TSENOR+ALPS | 2:8 | 40.27 | 44.21 | 57.42 | 59.85 | 29.94 | 35.02 | 21.16 | 49.96 | 53.07 | 25.80 | 61.41 | 42.03 |
| Bi-NM | 4:16 | 45.81 | 56.77 | 77.02 | 60.83 | 29.97 | 36.41 | 21.59 | 50.83 | 53.79 | 24.40 | 61.62 | 42.43 |
| TSENOR+ALPS | 4:16 | 35.66 | 37.49 | 51.27 | 61.21 | 32.74 | 36.24 | 22.10 | 54.14 | 52.71 | 28.20 | 62.11 | 43.68 |
| Bi-NM | 8:32 | 42.25 | 46.39 | 63.22 | 61.15 | 30.84 | 37.58 | 23.63 | 51.54 | 52.71 | 25.80 | 54.37 | 42.20 |
| TSENOR+ALPS | 8:32 | 33.42 | 33.25 | 47.26 | 62.13 | 33.90 | 37.75 | 22.78 | 51.54 | 53.43 | 27.00 | 62.17 | 43.84 |

Table 9: Performance analysis for (transposable) N:M pruning on LLaMA3.2-1B model. Lower values are preferred for perplexity, and higher values are preferred for zero-shot tasks.

| Method | Config | C4 ↓ | WT2 ↓ | PTB ↓ | PIQA ↑ | HS ↑ | ARC-E ↑ | ARC-C ↑ | WG ↑ | RTE ↑ | OQA ↑ | BoolQ ↑ | Avg ↑ |
|---|---|---|---|---|---|---|---|---|---|---|---|---|---|
| Bi-NM | 2:4 | 36.02 | 42.64 | 63.22 | 62.40 | 32.07 | 36.78 | 23.29 | 51.14 | 53.43 | 26.40 | 59.57 | 43.14 |
| TSENOR+ALPS | 2:4 | 30.69 | 36.88 | 52.06 | 61.97 | 32.02 | 37.42 | 23.63 | 51.46 | 53.79 | 27.00 | 53.76 | 42.63 |
| Bi-NM | 4:8 | 30.65 | 35.67 | 51.87 | 63.11 | 33.19 | 37.46 | 23.81 | 51.78 | 49.82 | 28.00 | 54.56 | 42.71 |
| TSENOR+ALPS | 4:8 | 27.76 | 31.95 | 44.43 | 63.76 | 33.09 | 37.25 | 22.78 | 51.54 | 53.79 | 28.00 | 59.91 | 43.77 |
| Bi-NM | 8:16 | 28.70 | 33.48 | 46.47 | 63.33 | 33.85 | 37.79 | 23.55 | 51.54 | 52.35 | 27.60 | 56.24 | 43.28 |
| TSENOR+ALPS | 8:16 | 26.28 | 30.33 | 40.71 | 63.60 | 34.08 | 39.27 | 24.40 | 49.33 | 53.43 | 28.60 | 57.40 | 43.76 |
| Bi-NM | 16:32 | 27.84 | 32.10 | 44.68 | 63.55 | 33.90 | 39.02 | 23.12 | 49.88 | 52.71 | 27.00 | 58.69 | 43.48 |
| TSENOR+ALPS | 16:32 | 25.57 | 28.38 | 39.30 | 63.28 | 34.41 | 39.31 | 23.21 | 52.01 | 54.87 | 28.20 | 60.12 | 44.43 |
| Bi-NM | 1:4 | 126.51 | 272.02 | 228.17 | 57.45 | 27.10 | 31.57 | 20.99 | 53.67 | 52.35 | 24.00 | 53.33 | 40.06 |
| TSENOR+ALPS | 1:4 | 72.35 | 122.32 | 137.06 | 56.96 | 26.99 | 31.61 | 21.33 | 49.64 | 51.26 | 23.80 | 55.41 | 39.63 |
| Bi-NM | 2:8 | 108.17 | 193.02 | 179.63 | 58.27 | 27.46 | 33.12 | 21.42 | 50.99 | 53.07 | 22.60 | 59.45 | 40.80 |
| TSENOR+ALPS | 2:8 | 56.41 | 85.11 | 99.08 | 58.49 | 27.38 | 33.59 | 20.99 | 51.62 | 53.07 | 25.00 | 60.00 | 41.27 |
| Bi-NM | 4:16 | 97.56 | 162.79 | 154.25 | 59.58 | 27.91 | 32.87 | 21.25 | 50.91 | 54.51 | 23.80 | 62.17 | 41.62 |
| TSENOR+ALPS | 4:16 | 48.74 | 67.69 | 87.67 | 58.76 | 28.41 | 33.84 | 21.42 | 52.41 | 53.07 | 24.80 | 62.23 | 41.87 |
| Bi-NM | 8:32 | 88.68 | 144.25 | 148.01 | 59.47 | 28.23 | 33.38 | 22.27 | 50.12 | 52.71 | 24.80 | 61.77 | 41.59 |
| TSENOR+ALPS | 8:32 | 44.41 | 63.34 | 82.29 | 59.09 | 28.82 | 33.84 | 22.70 | 50.67 | 51.26 | 25.40 | 61.83 | 41.70 |

Table 10: Performance analysis for (transposable) N:M pruning on OPT-350M model. Lower values are preferred for perplexity, and higher values are preferred for zero-shot tasks.

