# OpenReview forum: "TSENOR: Highly-Efficient Algorithm for Finding Transposable N:M Sparse Masks"
_NeurIPS.cc/2025/Conference — NeurIPS 2025 poster_

### Official Review · Reviewer_qhU9 · 2025-06-04

**Clarity:** 3
**Significance:** 2
**Originality:** 3
**Rating:** 5
**Confidence:** 4

**Summary:**

This paper proposes TSENOR, a novel and efficient algorithm for generating binary masks with transposable N:M sparsity, enabling acceleration in both forward and backward passes during neural network training. The authors formulate the mask generation problem as an entropy-regularized optimal transport problem and solve it using Dykstra’s algorithm. A GPU-optimized rounding step, composed of greedy selection and local search, ensures constraint satisfaction and mask quality. The method is integrated with several pruning frameworks (e.g., Wanda, SparseGPT, ALPS) and achieves substantial speedup (10–100×) and low reconstruction error (1–10%) on  LLaMA3.2–8B model.

**Questions:**

1. Why is Bi-NM used in Figure 5 as the primary comparison instead of 2-approx? In Figure 3, 2-approx shows better reconstruction error than Bi-NM, so it would be more convincing to include 2-approx as a baseline in Figure 5.

2. Have you considered GPU-accelerated implementations of 2-approx for fairer runtime benchmarking?
TSENOR is compared to 2-approx on CPU in Table 1. Leveraging NVIDIA’s cuOPT for general LP or other parallel kernels could offer a more apples-to-apples comparison with Network Flow.

**Ethical Concerns:**

["NO or VERY MINOR ethics concerns only"]

**Final Justification:**

After the rebuttle I decided to raise my score to 5

**Limitations:**

While they do briefly mention limitations in two sentences at the end of section 6, it's not enough. I'd encourage the authors to add a dedicated section on limitations. A thorough discussion of what's missing or where the current work falls short is truly invaluable for paving the way for impactful future research.

**Quality:**

3

**Strengths And Weaknesses:**

# Strengths

1. Reformulating the sparse mask search as an optimal transport problem is an insightful approach, grounded in theory and convex optimization.

2. Figure 3 shows that TSENOR achieves superior solution quality with significantly lower relative error compared to competing methods like 2-approx.

3. Table 1 and related experiments show significant acceleration due to GPU parallelism, making the method viable for large models.

4. Comprehensive empirical validation: Evaluation includes mask quality metrics, perplexity comparisons, and inference speedups on multiple sparsity levels.

# Weaknesses
1. Transposable N:M sparsity remains a relatively niche topic; thus, the broader significance and audience interest may be limited.

2. Alternative baselines could be enhanced: While the current comparisons are reasonable, further exploration of acceleration for classic heuristics like 2-approx on GPU may reinforce the case.

3. Incomplete code availability: The paper mentions GPU-efficient implementations, but only partial code appears to be provided. Without full implementation and integration examples, reproducibility and adoption may be hindered.

---

> ### Author Rebuttal · Authors · 2025-07-30
>
> We would like to thank the reviewer for their thoughtful feedback. Below, we provide some answers/clarifications.
>
> **Transposable N:M sparsity remains a relatively niche topic; thus, the broader significance and audience interest may be limited.**
>
> We appreciate this perspective. Transposable N:M sparsity addresses a critical challenge: accelerating both forward and backward passes during full fine-tuning without specialized hardware. As training costs become prohibitive, methods that enable efficient full fine-tuning on standard GPUs are essential. Our approach potentially enables researchers with limited resources to train and fine-tune large models more efficiently.
>
> **Alternative baselines could be enhanced: While the current comparisons are reasonable, further exploration of acceleration for classic heuristics like 2-approx on GPU may reinforce the case.**
>
> We appreciate this valuable suggestion. Due to the sequential nature of 2-approx (selecting the best feasible element at each step), direct GPU implementation is challenging. To our knowledge, the only available GPU implementation [1] uses CUDA kernels, but after contacting the authors, we found their code is incompatible with modern GPUs on our platform.
> We will consider implementing our own CUDA version of the 2-approx method as an enhanced baseline and compare it against TSENOR in both mask generation and fine-tuning settings in our revised manuscript.
>
> [1] Accelerating transformer pre-training with 2: 4 sparsity
>
> **Incomplete code availability: The paper mentions GPU-efficient implementations, but only partial code appears to be provided. Without full implementation and integration examples, reproducibility and adoption may be hindered.**
>
> Thank you for this valuable suggestion! We have prepared our complete implementation for public release. Due to conference guidelines prohibiting links in rebuttals, we will make the code available upon paper acceptance.
>
> **Why is Bi-NM used in Figure 5 as the primary comparison instead of 2-approx? In Figure 3, 2-approx shows better reconstruction error than Bi-NM, so it would be more convincing to include 2-approx as a baseline in Figure 5.**
>
> Bi-NM's advantage over 2-Approx and other methods only emerges during fine-tuning — it trains with non-transposable N:M masks and applies transposable masks only during gradient computation (backward pass) to achieve bidirectional acceleration. Therefore, we compare with Bi-NM in fine-tuning settings as it represents the strongest baseline.
> Since TSENOR generates transposable masks with either much higher quality or greater efficiency than 2-Approx and other competing methods (as shown in Figure 3 and Table 1), it predictably achieves superior fine-tuning performance. We will consider include fine-tuning comparisons with 2-Approx in our revised manuscript to provide more comprehensive evidence of TSENOR's advantages.
>
> **Have you considered GPU-accelerated implementations of 2-approx for fairer runtime benchmarking? TSENOR is compared to 2-approx on CPU in Table 1.**
>
> Please refer to our reply to Weakness 2.
>
> **Leveraging NVIDIA’s cuOPT for general LP or other parallel kernels could offer a more apples-to-apples comparison with Network Flow.**
>
> We appreciate this valuable suggestion. Currently, NVIDIA's cuOPT GitHub repository lacks documentation for its Python API, making direct comparison challenging. We have included comparisons in Table 1 with cuPDLP — a GPU-accelerated PDLP solver for linear programming, which offers similar insights (cuOPT also employs GPU-accelerated PDLP methods for solving LP problems). We will make our best effort to include a comparison with NVIDIA's cuOPT to serve as a stronger baseline for Network Flow methods in our revised manuscript.
>
> **While they do briefly mention limitations in two sentences at the end of section 6, it's not enough. I'd encourage the authors to add a dedicated section on limitations. A thorough discussion of what's missing or where the current work falls short is truly invaluable for paving the way for impactful future research.**
>
> Thank you for this valuable suggestion! We will add a new section in our revised manuscript to thoroughly discuss the current constraints of our approach and directions for future improvements.

---

### Official Review · Reviewer_NiHC · 2025-06-23

**Clarity:** 2
**Significance:** 2
**Originality:** 3
**Rating:** 4
**Confidence:** 3

**Summary:**

The paper introduces TSENOR, an algorithmic framework for efficiently generating transposable N:M sparse masks, essential for accelerating both forward and backward passes in neural network training. The key idea is formulating the mask generation problem as multiple optimal transport problems with entropy regularization, solved through Dykstra’s algorithm followed by a specialized rounding procedure. TSENOR is integrated into popular pruning methods such as Wanda, SparseGPT, and ALPS. The authors report significant computational speedups and reduced error relative to existing heuristics. Experimental results on large-scale models such as LLaMA demonstrate the practical efficacy of TSENOR, especially highlighting performance gains at larger M-values (e.g., 16:32 sparsity).

**Questions:**

1. Clarification of Computational Benefits at Larger M-values: Your claim— "Our experiments show that networks with larger M values (e.g., 16:32) provide the same computational benefits as smaller M values but with much less impact on model accuracy"— is unclear.
    * Which specific experiments support this claim?
    * Given that larger M-values inherently become less structured, it is counterintuitive that computational benefits remain constant. Could you clarify how these masks remain efficiently acceleratable?
2. Accuracy-Speed Trade-off Analysis: If I understand correctly, transposable N:M sparsity should accelerate both training and inference. However, the current manuscript lacks explicit results that clearly illustrate improved accuracy-speed trade-offs.
    * Could you clarify if and where such comparative results exist in the paper? If not, can you provide a rationale or consider including this comparison?

**Ethical Concerns:**

["NO or VERY MINOR ethics concerns only"]

**Final Justification:**

Thank you for the detailed explanation.
The rebuttal response solved my main concerns about the large M (I misunderstood this part, especially about wall-clock latency).
And I now understand the significance of the proposed entropy regularization.
With the clarification, I understands the proposed methods also has practical speed gains.
After the rebuttal all l my concerns have been resolved.
Now I have raised my rating.

**Limitations:**

yes

**Quality:**

3

**Strengths And Weaknesses:**

## Strengths:
1. Novel Optimization Formulation: Formulating mask selection as an optimal transport problem with entropy regularization is both elegant and theoretically appealing, significantly simplifying computational complexity compared to previous methods.
2. Efficiency and Scalability: The GPU-optimized implementation of TSENOR demonstrates remarkable efficiency and scalability, achieving substantial speedups (up to 100×) over baseline methods, enabling its practical application to large-scale language models.
3. Practical Integration with Popular Methods: The proposed method seamlessly integrates with existing pruning frameworks like Wanda, SparseGPT, and ALPS, allowing straightforward adoption within the broader neural network pruning community.
4. Comprehensive Experimental Validation: Experiments are thorough and extensive, covering both single-matrix evaluations and full-scale pruning scenarios on LLaMA models, clearly demonstrating computational advantages and acceptable accuracy trade-offs.

## Weaknesses:
1. Clarity on Practical Gain: While the method clearly demonstrates computational efficiency, the practical gain in terms of the speed-accuracy trade-off in training and inference is not fully explicit. Specifically, results explicitly comparing accuracy versus computational speed between transposable and non-transposable N:M masks (or between regular dense matmul) are limited or unclear.
2. Technical Novelty Moderation: Although the connection to optimal transport is interesting and practically beneficial, the overall approach leverages well-established optimization techniques (e.g., entropy regularization, Dykstra's algorithm) and therefore offers moderate conceptual novelty.
3. Limited Detailed Discussion on Larger M Values: The claim regarding the computational benefits and accuracy at larger M-values (e.g., 16:32) seems insufficiently justified. It's unclear how these larger M-values maintain similar computational benefits as smaller structured masks, considering the tendency towards unstructured patterns at larger M.

## Overall Evaluation:
The paper presents an interesting approach by leveraging optimal transport theory for efficient mask generation in neural network pruning, particularly relevant given the increasing prominence of large language models. While computationally impressive, the practical advantage regarding accuracy-speed trade-offs is not fully articulated. The technical novelty is moderate, considering the method largely relies on known optimization techniques.
Thus, I lean towards a weak rejection given the current state, primarily because the critical practical gains (accuracy-speed trade-off improvements) are not thoroughly demonstrated. The paper could significantly benefit from narrowing its scope, providing more explicit comparative results, and further justifying claims regarding large M-value effectiveness.

---

> ### Author Rebuttal · Authors · 2025-07-30
>
> We would like to thank the reviewer for their thoughtful feedback. Below, we provide some answers/clarifications.
>
> **Clarity on Practical Gain: While the method clearly demonstrates computational efficiency, the practical gain in terms of the speed-accuracy trade-off in training and inference is not fully explicit. Specifically, results explicitly comparing accuracy versus computational speed between transposable and non-transposable N:M masks (or between regular dense matmul) are limited or unclear.**
>
> We respectfully disagree with the reviewer's assessment, as we believe Section 5.2.1 already provides explicit evidence demonstrating the superior speed-accuracy trade-off achieved by transposable N:M sparsity with large M values.
> Figure 4 (upper) shows that when pruning LLaMA3.2-8B, the performance gap between transposable and standard N:M sparsity diminishes dramatically as M increases from 4 to 32, with transposable N:M performance growing rapidly. Figure 4 (lower) demonstrates that transposable N:M achieves significantly higher computational speedup than standard N:M during training. Moreover, the speedup of transposable N:M remains consistent across different M values. Together, these figures show that transposable N:M sparsity with large M values (M=32) achieves comparable accuracy to standard N:M while maintaining significant computational speedup during training.
>
> **Technical Novelty Moderation: Although the connection to optimal transport is interesting and practically beneficial, the overall approach leverages well-established optimization techniques (e.g., entropy regularization, Dykstra's algorithm) and therefore offers moderate conceptual novelty.**
>
> We respectfully disagree with the reviewer's assessment. Our paper presents several technical novelties:
> + In Section 3.2, we introduce a novel approach based on entropy regularization that solves masks with arbitrary transposable N:M sparsity. The key innovation is utilizing GPU tensor operations to solve millions of blocks simultaneously, making our approach highly parallelizable.
> + In Section 3.3, we propose a novel rounding procedure that converts fractional solutions to high-quality binary masks through greedy selection and local search. Our tensor-based implementation processes millions of blocks simultaneously, achieving up to 10³× speedup over CPU implementation.
> + In Section 4, we integrate TSENOR with existing layer-wise N:M pruning frameworks (Wanda, SparseGPT, ALPS) to generate transposable N:M sparse networks. For ALPS, we provide novel convergence guarantees for the resulting framework.
>
> We believe these contributions represent algorithmic and theoretical advances beyond simply applying existing optimization techniques.
>
> **Limited Detailed Discussion on Larger M Values: The claim regarding the computational benefits and accuracy at larger M-values (e.g., 16:32) seems insufficiently justified. It's unclear how these larger M-values maintain similar computational benefits as smaller structured masks, considering the tendency towards unstructured patterns at larger M.**
>
> We have provided comprehensive experimental evidence demonstrating the benefits of larger M-values — please refer to our detailed response to Weakness 1.
>
> **Clarification of Computational Benefits at Larger M-values: Your claim— "Our experiments show that networks with larger M values (e.g., 16:32) provide the same computational benefits as smaller M values but with much less impact on model accuracy"— is unclear. Which specific experiments support this claim?**
>
> The experimental evidence supporting this claim is presented in Figure 4 and discussed in Section 5.2.1 — please refer to our detailed response to Weakness 1.
>
>
> **Given that larger M-values inherently become less structured, it is counterintuitive that computational benefits remain constant. Could you clarify how these masks remain efficiently acceleratable?**
>
> The speedup is achieved using the NM-SpMM CUDA kernel [1], which employs memory access optimization and hierarchical blocking mechanisms for sparsity-aware acceleration. These optimizations enable near-theoretical peak performance across different N:M sparsity levels, ensuring that computational benefits remain nearly constant regardless of M value.
>
> [1] NM-SpMM: Accelerating Matrix Multiplication Using N:M Sparsity with GPGPU
>
> **Accuracy-Speed Trade-off Analysis: If I understand correctly, transposable N:M sparsity should accelerate both training and inference. However, the current manuscript lacks explicit results that clearly illustrate improved accuracy-speed trade-offs. Could you clarify if and where such comparative results exist in the paper? If not, can you provide a rationale or consider including this comparison?**
>
> In Figure 4, we demonstrate that transposable N:M sparsity maintains comparable accuracy to standard N:M sparsity (upper figure) while achieving 30% to 4× speedup during training (lower figure) — please refer to our detailed response to Weakness 1.

---

### Official Review · Reviewer_xTpm · 2025-06-27

**Clarity:** 4
**Significance:** 3
**Originality:** 2
**Rating:** 5
**Confidence:** 3

**Summary:**

TSENOR is a work providing high quality N:M masks that scales to large models with efficient algorithms and exploitation of GPUs speedup in tensor multiplication. TSENOR generates weight mask by adapting Dykstra’s algorithm to solve an optimal transport problem. A greedy rounding method is further applied to create transposable binary mask.

**Questions:**

If one cares only about inference, can TSENOR be easily adapted to provide non-transposable masks?

It seems that optimal solution is more reachable when M is smaller than 8. Is it a general conclusion that larger M is always a significant improvements in solution quality (as shown in Upper Figure 4) in different types of networks?

**Ethical Concerns:**

["NO or VERY MINOR ethics concerns only"]

**Final Justification:**

I find this paper well written and the idea is noval. The authors replied to my questions with good answers and the discussion with other reviewers seems effective. I have no additional opinions about this paper, and I find my initial score the most suitable for this paper.

**Limitations:**

Yes.

**Paper Formatting Concerns:**

No.

**Quality:**

3

**Strengths And Weaknesses:**

Strengths:

TSENOR well balances the effectiveness and efficiency of N:M sparsity, which enables large M for higher quality solutions while keeping solving time within a feasible range.
The paper is well-written, algorithms are motivated in a clear way and explained with good examples.
Ablation study in the appendix covered most concerns of mine. For different features of TSENOR, there are evidence to justify their usage.

Weaknesses:

For exploiting the GPU tensor multiplication, it would be nice to have a usable framework for TSENOR.
Source code is not provided, and it is difficult for anyone tends to reproduce the results even though hyperparameters are provided.
Fine-tuning performance of TSENOR is not very well presented.

---

> ### Author Rebuttal · Authors · 2025-07-30
>
> We would like to thank the reviewer for their thoughtful feedback. Below, we provide some answers/clarifications.
>
> **For exploiting the GPU tensor multiplication, it would be nice to have a usable framework for TSENOR. Source code is not provided, and it is difficult for anyone tends to reproduce the results even though hyperparameters are provided.**
>
> Thank you for this valuable suggestion! We have prepared our complete implementation for public release. Due to conference guidelines prohibiting links in rebuttals, we will make the code available upon paper acceptance.
>
> **Fine-tuning performance of TSENOR is not very well presented.**
>
> We appreciate the reviewer's valuable suggestion! We will include more fine-tuning experiments comparing TSENOR with other transposable methods in our revised manuscript. We have conducted additional experiments comparing TSENOR with Bi-NM on OPT-1.3B, as shown in the following table (- denotes still running—will be updated later):
>
> |N:M |2:4 | 4:8 | 8:16 | 16:32|
> |---|---|---|---|---|
> |TSENOR|66.2| 46.7| 38.4|35.8|
> |Bi-NM |67.9|43.6|-|40.2|
>
>
> **If one cares only about inference, can TSENOR be easily adapted to provide non-transposable masks?**
>
> This is a good point. TSENOR is designed for generating transposable masks, but can be easily adapted to generate non-transposable masks. In this case, the per-block problem in Equation (2) only involves row-wise N:M constraints and can be solved directly by selecting $S_{ij}=1$ if $j$ is among the largest N values in row $i$. This version can still be integrated with layer-wise pruning approaches, including Wanda, SparseGPT, and ALPS, as discussed in Section 4.
>
> **It seems that optimal solution is more reachable when M is smaller than 8. Is it a general conclusion that larger M is always a significant improvements in solution quality (as shown in Upper Figure 4) in different types of networks?**
>
> Yes, this is a general conclusion. Transposable N:M sparsity with smaller M imposes much stricter constraints — for example, an 8×8 block with 2:4 transposable sparsity has about 65 million feasible masks, while 4:8 transposable sparsity has greater than 100 billion feasible masks. Larger M values provide more flexibility in mask selection and therefore consistently yield better solution quality across different network types. This is further validated by our results on OPT-1.3B (as shown above), where larger M values consistently achieve higher performance.

---

> > ### Comment · Reviewer_xTpm · 2025-08-04
> >
> > Thank you for your explanations. I will maintain my score.

---

> > > ### Author Response · Authors · 2025-08-06
> > >
> > > We sincerely appreciate the time you took to review our work and are glad that our revisions addressed your concerns.

---

### Official Review · Reviewer_oDcR · 2025-07-04

**Clarity:** 3
**Significance:** 2
**Originality:** 3
**Rating:** 5
**Confidence:** 4

**Summary:**

This paper presents TSENOR, an efficient algorithm for generating transposable N:M sparse masks that enable both forward and backward pass acceleration in large neural networks like LLMs. TSENOR uses an entropy-regularized optimal transport formulation solved with GPU-parallelizable Bregman projections to find high-quality fractional masks, then refines them into valid binary masks with a fast rounding procedure. Integrated with popular pruning frameworks like SparseGPT, Wandda and ALPS, TSENOR scales to billion-parameter models, supports flexible N:M ratios, and achieves substantial speedup with minimal performance loss. This work has showed it can achieve siganifically less relative error compared to other transposable pruning methods. Furthermore, it integrates with ALPS and achieves comparable performance with non-transposable ones on LLMs like LLaMA3.

**Questions:**

1. In section 5.2.1, it evaluates the tradeoff between speedup and performance loss, which seems to be one-shot pruning without fine-tuning.  If I understand correctly, it compares the one-shot pruning performance and estimates the speedup of fine-tuning. A more straightforward way might be to evaluate both on fine-tuning settings.
2. What are the fine-tuning settings? If fine-tuning hard, will the transposable method like Bi-NM also work very well?

**Ethical Concerns:**

["NO or VERY MINOR ethics concerns only"]

**Final Justification:**

My concerns are cleared by the experiments provided in the rebuttal and I decide to raise my score.

**Limitations:**

My main concern is that although the paper proposes an efficient method for generating transposable N: M sparse masks, the practical benefit may be limited if, after fine-tuning, models perform similarly to those produced by other transposable sparsity methods.

**Quality:**

3

**Strengths And Weaknesses:**

Strengths:
1. The paper has formulated transposable N: M sparsity cleanly as an entropy-regularized optimal transport problem with capacity constraints, which is mathematically well-founded.
2. Dykstra’s algorithm and the proposed rounding algorithm with the local search are highly efficient, which can achieve the lowest relative error compared to the optimal solution while speeding it up by 100 times.
3. This method looks like an effective plug-in for the ALPS method. And for one-shot pruning, this transposable N: M sparsity can achieve comparable performance with non-transposable ones on LLMs.

Weaknesses:
1. As transposable N: M sparsity is designed to speed up both forward and backward, it is best to show practical benefit on more training or fine-tuning settings. While this paper has demonstrated it by fine-tuning LLaMA3.2-8B, although with much lower relative error compared to other transposable methods, the final performance improvement over Bi-NM seems limited.

---

> ### Author Rebuttal · Authors · 2025-07-31
>
> We would like to thank the reviewer for their thoughtful feedback. Below, we provide some answers/clarifications.
>
> **As transposable N: M sparsity is designed to speed up both forward and backward, it is best to show practical benefit on more training or fine-tuning settings.**
>
> We appreciate the reviewer's valuable suggestion. We have conducted additional fine-tuning experiments comparing TSENOR with Bi-NM on OPT-1.3B, with PTB perplexity results shown below (- denotes still running—will be updated later):
>
> |N:M |2:4 | 4:8 | 8:16 | 16:32|
> |---|---|---|---|---|
> |TSENOR|66.2| 46.7| 38.4|35.8|
> |Bi-NM |67.9|43.6|-|40.2|
>
> As shown, TSENOR outperforms Bi-NM, particularly at larger M values. We will include more fine-tuning experiments across multiple models in our revised manuscript to further demonstrate TSENOR's practical benefits in fine-tuning settings.
>
>
> **While this paper has demonstrated it by fine-tuning LLaMA3.2-8B, although with much lower relative error compared to other transposable methods, the final performance improvement over Bi-NM seems limited.**
>
> Fine-tuning is a highly effective step for performance recovery, and performance gaps between different pruning methods narrow significantly after fine-tuning [1,2]. Therefore, TSENOR's 3% improvement over Bi-NM after fine-tuning demonstrates TSENOR's superiority as a pruning algorithm, as these gains are achieved within an extremely competitive performance range.
>
> Additionally, TSENOR reduces the memory footprint during fine-tuning. While Bi-NM requires storing two different N:M sparse masks (one for forward and one for backward passes), TSENOR needs only a single transposable N:M sparse mask.
>
>
> [1] Sparse Matrix in Large Language Model Fine-tuning
>
> [2] A Simple and Effective Pruning Approach for Large Language Models
>
> **What are the fine-tuning settings? If fine-tuning hard, will the transposable method like Bi-NM also work very well?**
>
> The fine-tuning settings are provided in Appendix B.1. For both Bi-NM and TSENOR, we fine-tuned them on the first shard of the C4 dataset, which includes over 150 million tokens. We terminated fine-tuning until the loss showed no significant decrease. Bi-NM achieves backward speedup by using approximate gradients during fine-tuning. This approximation inherently limits its fine-tuning performance improvement compared to TSENOR, which maintains exact gradients throughout training.
>
> **In section 5.2.1, it evaluates the tradeoff between speedup and performance loss, which seems to be one-shot pruning without fine-tuning. If I understand correctly, it compares the one-shot pruning performance and estimates the speedup of fine-tuning. A more straightforward way might be to evaluate both
> on fine-tuning settings.**
>
> Thank you for this valuable suggestion! We compare fine-tuning performance for transposable and standard N:M sparsity across different N:M patterns on LLaMA3.2-1B in the following table. We will include this comparison in our revised manuscript.
>
> |N:M |2:4 | 4:8 | 8:16 | 16:32|
> |---|---|---|---|---|
> |Transposable N:M|22.1| 19.7| 18.4|17.7|
> |Standard N:M |19.4|18.1|17.4|17.1|
>
>
>
> **My main concern is that although the paper proposes an efficient method for generating transposable N: M sparse masks, the practical benefit may be limited if, after fine-tuning, models perform similarly to those produced by other transposable sparsity methods.**
>
> Since TSENOR generates pruned models with much higher performance compared to other transposable sparsity methods after one-shot pruning, it provides advantages in two key scenarios: (i) Limited fine-tuning: When training resources are constrained, TENSOR maintains superior performance over other methods due to its stronger starting point. (ii) Early termination: TENSOR can be early terminated during fine-tuning while achieving comparable results to fully fine-tuned models from other transposable methods, thereby saving computational resources.
>
> Even with full fine-tuning, TSENOR maintains a non-negligible improvement over Bi-NM, as mentioned previously.

---

> > ### Author Response · Authors · 2025-08-04
> >
> > Updated fine-tuning results comparing TSENOR with Bi-NM on OPT-1.3B (PTB perplexity):
> > |N:M |2:4 | 4:8 | 8:16 | 16:32|
> > |---|---|---|---|---|
> > |TSENOR|66.2| 46.7| 38.4|35.8|
> > |Bi-NM |67.9|43.6|42.2|40.2|

---

### Note · Authors · 2025-08-13

Dear AC,

We would like to provide a summary following the discussion phase.

During discussions, Reviewer qhU9 [score 4] expressed appreciation for our technical/experimental contributions. In response to their request for additional experiments, we have provided partial results on N:M patterns supported by AWS Trainium and Inferentia (with more experiments ongoing) on LLaMA3-8B, which continue to demonstrate TSENOR's superior performance. We have also prepared our complete implementation and will make the code publicly available upon acceptance, addressing reproducibility concerns.

Reviewer xTpm appears satisfied with our responses and maintains strong support.

We believe we have addressed all concerns from Reviewers oDcR [score 4] and NiHC [score 3], though we have not received further comments. Notably, for Reviewer NiHC's main concern about the lack of speed-accuracy trade-off analysis comparing transposable and non-transposable N:M masks during training, our detailed response referencing Figure 4 and Section 5.2.1 demonstrates that we already provide comprehensive evidence of this trade-off in the manuscript.

We remain committed to incorporating all reviewer suggestions in our revision and appreciate the constructive feedback that has strengthened our work.

---

### Decision · Program_Chairs · 2025-09-17

**Decision:**

Accept (poster)

**Comment:**

This paper introduces TSENOR, a scalable and GPU-efficient algorithm for generating transposable N:M sparse masks, which accelerate both forward and backward passes in large-scale neural network training. The method formulates mask generation as an entropy-regularized optimal transport problem, solved with Dykstra’s algorithm and followed by a fast rounding procedure. Experiments on LLaMA and OPT models show up to 100× speedup over existing methods, improved reconstruction error, and performance competitive with or better than Bi-NM, especially at larger M values (e.g., 16:32).

Reviewers appreciate the clean formulation, strong efficiency gains, and thorough empirical validation, as well as the practical integration with existing one-shot pruning frameworks. Main concerns included the clarity of fine-tuning comparisons, the significance of larger M values, moderate conceptual novelty, and reproducibility due to initially missing full code release.

The rebuttal addressed these issues with additional fine-tuning results, clarifications on computational benefits at larger M, and a commitment to releasing full code and adding a limitations section. Multiple reviewers raised their scores after discussion, leading to a positive consensus. Overall, this is a solid and well-executed contribution that advances the state of the art in structured sparsity.